# HLA class I peptide polymorphisms contribute to class II DQβ0603:DQα0103 antibody specificity

N. Remi Shih[1,8], Thoa Nong[1,8], Cathi Murphey[2], Mayra Lopez-Cepero[1], Peter W. Nickerson[3], Jean-luc Taupin[4], Magali Devriese[4], Jakob Nilsson[5], Marie-Benedicte Matignon[6], Robert A. Bray[7,9] & Jar-How Lee[1,9]✉

Antibodies reactive to human leukocyte antigens (HLA) represent a barrier for patients awaiting transplantation. Based on reactivity patterns in single-antigen bead (SAB) assays, various epitope matching algorithms have been proposed to improve transplant outcomes. However, some antibody reactivities cannot be explained by amino acid motifs, leading to uncertainty about their clinical relevance. Antibodies against the HLA class II molecule, DQβ0603:DQα0103, present in some candidates, represent one such example. Here, we show that peptides derived from amino acids 119-148 of the HLA class I heavy chain are bound to DQβ0603:DQα0103 proteins and contribute to antibody reactivity through an HLA-DM-dependent process. Moreover, antibody reactivity is impacted by the specific amino acid sequence presented. In summary, we demonstrate that polymorphic HLA class I peptides, bound to HLA class II proteins, can directly or indirectly be part of the antibody binding epitope. Our findings have potential important implications for the field of transplant immunology and for our understanding of adaptive immunity.

The presence of donor-specific HLA antibody (DSA) has represented a challenge to solid organ transplantation since it was first demonstrated in 1969 with the cytotoxicity crossmatch[1]. Over the years, various cell-based assays, with improved sensitivity, were introduced to improve outcomes[2,3]. When a multiplex, bead-based assay that used single HLA antigens was introduced in 2003[4], the HLA antibody profile of patients could be clearly determined even when patients possessed a complex combination of HLA antibodies. The SAB assay has now become an integral part of determining donor-recipient compatibility and has resulted in significant changes to the allocation algorithm for solid organ transplantation[5]. In the United States, UNOS (United Network for Organ Sharing) has established the calculated panel reactive antibody (cPRA). The cPRA is based on the recipient's HLA antibody profile derived from solid phase assays and the frequency of the selected HLA antigens in a donor population. The cPRA has become an important factor used in renal organ allocation[5] in the United States. In addition to cPRA, refinements to HLA antibody assessments have permitted the concept of the virtual crossmatch (VXM). The VXM has become common practice at many centers and has been utilized in lieu of a physical cell-based crossmatch[6]. The viability of both the cPRA and VXM are dependent on accurate HLA antibody assignments obtained from SAB assays. However, over the years, many publications have established that SAB assays can exhibit discordant results when compared to cell-based crossmatch methods[7]. Studies have demonstrated that denatured HLA antigens on the beads can expose cryptic epitopes that may explain some discordant results[8,9]. Nevertheless, there are other

[1]Terasaki Innovation Center, Los Angeles, CA, USA. [2]Southwest Immunodiagnostics, Inc., San Antonio, TX, USA. [3]Max Rady College of Medicine, University of Manitoba, Winnipeg, MB, Canada. [4]Laboratoire d'Immunologie et Histocompatibilité and INSERM U976 IRSL, Hôpital Saint-Louis APHP, Paris, France. [5]Department of Immunology, University Hospital Zurich, Zurich, Switzerland. [6]Department of Nephrology and Transplantation, Hôpital Henri Mondor APHP, Creteil, France. [7]Department of Pathology, Emory University, Atlanta, GA, USA. [8]These authors contributed equally: N. Remi Shih, Thoa Nong. [9]These authors jointly supervised this work: Robert A. Bray, Jar-How Lee. ✉e-mail: jarhowlee@tictx.org

discordant results, especially for class II antibody reactivity, that are not completely understood[7]. Determining the cause of these discordant antibody reactivities can facilitate the interpretation of a patient's antibody profile and thereby improve the accuracy of the cPRA and VXM.

Alloantibodies, directed against the HLA DQ molecule (α and β chains) have been frequently observed in transplant recipients[10]. De novo, donor-specific HLA-DQ antibodies have been shown to be the most prevalent HLA antibody formed, post-transplant, in DSA-positive recipients and has been significantly associated with chronic antibody-mediated rejection (AMR) and lower overall graft survival[11]. However, the observations that DQ antibodies can be frequently detected in non-sensitized patients and may not produce a positive cell-based XM have generated confusion and concern related to the interpretation of results and how best to utilize the information for patient management[7]. One explanation for discordance between cell-based and bead-based assays is the fact that HLA antigen bearing beads display comparable amounts of DR and DQ molecules while cells express significant lower amounts of DQ antigens on their surface compared to DR[12]. This difference leads to increased sensitivity of bead-based assays for detecting HLA DQ antibodies. However, this fact cannot explain why non-sensitized patients produced such HLA DQ-reactive antibodies. Thus, investigating the formation and the nature of DQ antibody production could provide useful information for solid organ allocation and AMR treatment.

The loading and binding of peptides to class II molecules is controlled through an HLA DM-dependent process via the removal of the class II invariant chain peptide (CLIP) from the peptide-binding groove of class II proteins[13]. This action facilitates the loading of alternative peptides derived from internalized or endogenous proteins. It has been reported that the HLA-DR3 protein expressed in the B cell line 721.174 (a cell line which has deletions from DRA to DPB1 in both copies of the chromosomes) failed to react with a DR3 monoclonal antibody[14]. However, transfection of HLA DMA and DMB genes into 721.174(DR3) cells restored the reactivity of the monoclonal antibody with the DR3 protein[15]. Similar observations have been reported with certain anti-DQ monoclonal antibodies failed to recognize the DQ proteins expressed on either mouse L cell or human HeLa cell transfectants most likely due to the lack of appropriate peptides bound to the binding groove of the DQ protein[16]. Patient sera that react to the DQβ0603:DQα0103 heterodimer without cross-reacting with other HLA DQ antigens are frequently observed. This specific and unique reactivity cannot be

explained by either unique or shared protein sequences among other DQ antigens. Given that no obvious amino acid sequence motif accounts for the reactivity, we hypothesized that DQβ0603:DQα0103 specific sera might be interacting with a DQβ0603:DQα0103/peptide complex through a similar peptide-dependent mechanism.

In this study, we examine twenty-two patient sera with DQβ0603:DQα0103 reactivity and report that polymorphic peptides, either inter-locus or intra-locus, derived from the HLA class I heavy chain are loaded into the DQβ0603:DQα0103 peptide-binding groove through an HLA-DM-dependent process to form the epitope recognized by DQβ0603:DQα0103-specific antibodies. Furthermore, the polymorphism of the peptide sequences bound to the groove of the protein alters the avidity of the antibodies to the DQβ0603:DQα0103 protein. Clearly, the uniqueness of this kind of antibody reactivity warrants further investigation.

## Results

### DQβ0603:DQα0103 positive sera have different levels of serological reactivity to DQβ0603:DQα0103 proteins extracted from different EBV transformed B cell lines

Six DQβ0603:DQα0103 specific sera (S1-S6) were identified from patients via routine HLA antibody monitoring. These six sera reacted only to the DQβ0603:DQα0103 antigen bead. All other DQ antigen beads on the class II SAB panel were negative. These six sera were then tested against a bead panel of DQβ0603:DQα0103 proteins purified from five homozygous EBV transformed B cell lines to confirm the presence of DQβ0603:DQα0103 antibody reactivity. All five B cell lines possessed different class I genotypes (Table 1). Their serological reactivity is shown in Fig. 1a. Surprisingly, the reactivity of the six DQβ0603:DQα0103 specific sera showed at least two distinct profiles to those five DQβ0603:DQα0103 proteins. In contrast, four monoclonal antibodies, among which three recognizing DQα and one recognizing DQβ, and two DQ6 positive control allo-sera (PC #1 and PC #2) showed similar reactivity to the DQβ0603:DQα0103 antigen bearing beads demonstrated that these five DQβ0603:DQα0103 beads have a similar amount of protein on each bead (Fig. 1b).

One of the six sera, S1, showed stronger reactivity to the DQβ0603:DQα0103 proteins derived from the 9058 and 9062 cell lines which are both HLA A*02:01. The other three cell lines, 9060, 9065 and 9105, are non-A*02:01. The other five sera showed reaction patterns to the five DQβ0603:DQα0103 antigens with slight but distinguishable variations in the relative strength. The significantly different reactivity of S1 to the same protein derived from five different B cell lines indicated that post-translational influence from the host cell lines was likely responsible for the variation.

### DQβ0603:DQα0103 positive sera reacted to recombinant DQβ0603:DQα0103 proteins derived from transfectants of CD74 and DM positive host cell lines

Since S1 preferentially reacted to the DQβ0603:DQα0103 protein isolated from the two HLA-A*02:01 positive B cell lines, the involvement of HLA class I expression was investigated. The antibody reactivity of the six DQβ0603:DQα0103 sera was measured against recombinant DQβ0603:DQα0103 proteins extracted from six stable transfectants via the LABScreen SAB assay protocol (Fig. 2a). Importantly, each of the six host cell lines possessed a different genetic background that affects the expression of invariant chain (CD74), HLA-DM or HLA class I, specifically A*02:01 (Table 1). The same four monoclonal antibodies and two positive control sera were used to assess the amount and the integrity of the recombinant DQβ0603:DQα0103 proteins on the beads. The protein isolated from the CD74 −, DM− cell line, K562, has lower reactivity to all four monoclonal antibodies and PC#2 (Fig. 2b) when compared to proteins isolated from DM- BLS or T2 suggesting that CD74 was required to stabilize the DQ protein.

**Table 1 | The HLA class I genotypes of the EBV-transformed B cells and transfectant hosts**

| Cell | HLA class I type | | | Expression | | | |
|---|---|---|---|---|---|---|---|
| | | | | class I | class II | CD74 | DM |
| 9058 | A*02:01 | B*45:01 | C*16:01 | + | + | + | + |
| 9060 | A*01:01 | B*15:01 | C*03:03 | + | + | + | + |
| 9062 | A*02:01 | B*38:01 | C*12:03 | + | + | + | + |
| 9065 | A*03:01 | B*07:02 | C*07:02 | + | + | + | + |
| 9105 | A*01:01 | B*35:02 | C*04:01 | + | + | + | + |
| LCLKO | − | − | C*01:02 | ±[a] | − | + | + |
| LCL3023 | A*02:01 | B*51:01 | C*01:02 | + | − | + | + |
| T2 | A*02:01 | B*51:01 | C*01:02 | + | − | + | − |
| T2DM | A*02:01 | B*51:01 | C*01:02 | + | − | + | + |
| K562 | A*11:01 | B*18:01 | C*03:04 | − | − | − | − |
| | A*31:01 | B*40:01 | C*05:01 | | | | |
| BLS | A*11:01 | B*41:02 | C*02:02 | + | − | + | − |
| | A*66:01 | B*51:01 | C*17:03 | | | | |

[a]Very low level of HLA class I protein expression on cell surface.

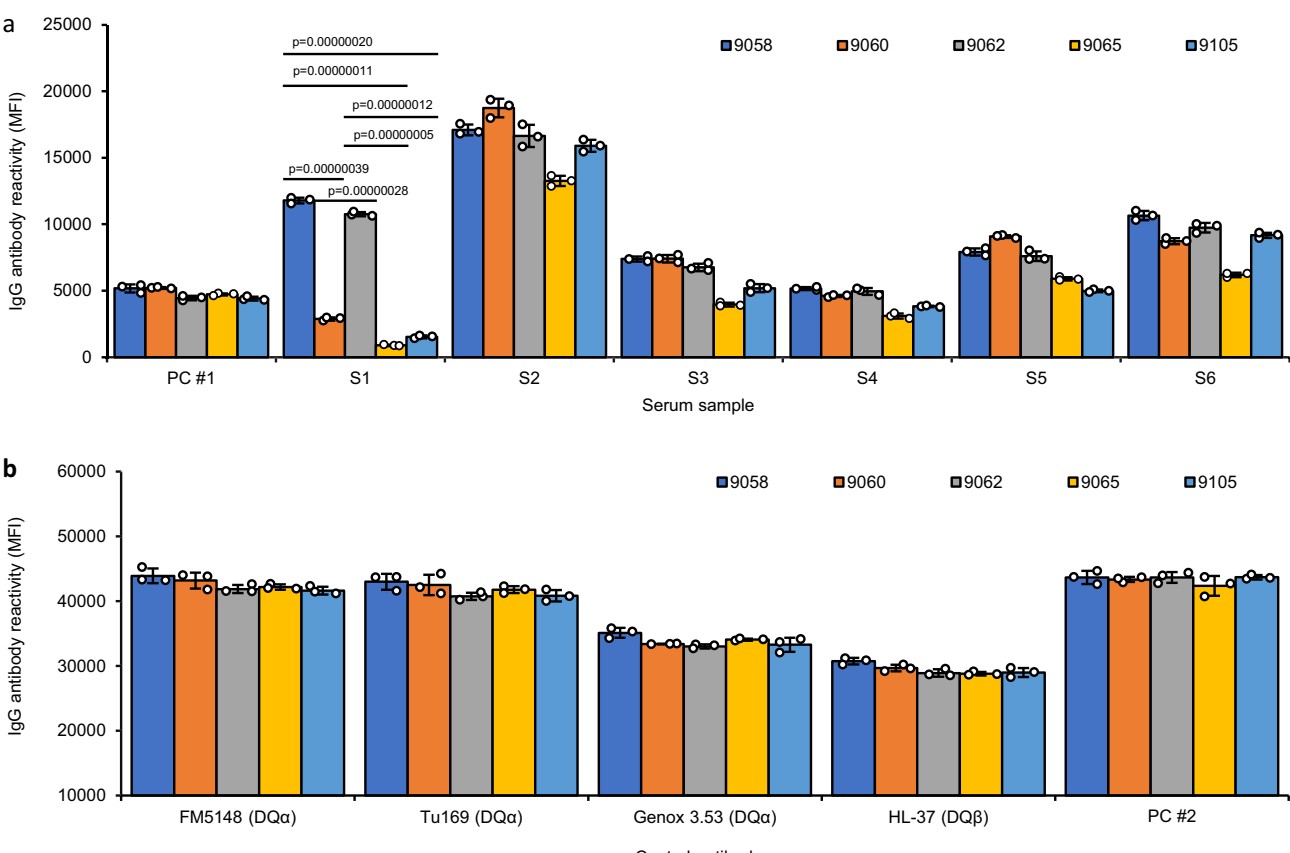

**Fig. 1 | Reactions of DQβ0603:DQα0103 specific sera to the DQβ0603:DQα0103 protein purified from five different B cell lines.** DQβ0603:DQα0103 protein purified from five different B cell lines, 9058 (dark blue), 9060 (orange), 9062 (gray), 9065 (yellow) and 9105 (light blue) were tested against eight human sera and four mouse monoclonal antibodies. Antibody reactivity was measured as mean fluorescence intensity (MFI) using the LABScreen SAB assay protocol. Data are presented as mean values +/- SD. Bars indicate the mean, whiskers indicate the standard deviation, and white dots with black circle represent the individual data points from triplicate tests ($n = 3$). **a** With six DQβ0603:DQα0103 specific sera and a weak positive control serum, PC #1. **b** With four monoclonal antibodies. FM5148 and Tu169 are specific for a monomorphic epitope on DQα, Genox 3.53 is specific for DQα01 and HL-37 is specific for DQβ01,03 and a strong positive control serum, PC #2. Statistical significance was determined using one-way ANOVA followed by Bonferroni post hoc test. The complete statistical analysis results are provided in Source data file. Only results with $p < 0.000001$ are shown.

Five of the six sera, (S1-S5) showed positive reactivity against beads bearing recombinant DQβ0603:DQα0103 protein from T2DM host, but negative on the recombinant DQβ0603:DQα0103 protein from T2 host (Fig. 2a). Since T2 and T2DM differ only on the expression of DM in T2DM, this result would indicate that DM expression is essential for the assembly of the epitope recognized by these DQβ0603:DQα0103 sera. Negative reactions on the recombinant DQβ0603:DQα0103 proteins derived from BLS (CD74 + , DM − ) or K562 (CD74 − , DM − ) hosts and positive reaction on the recombinant DQβ0603:DQα0103 protein derived from LCL3023(CD74 + ,DM + ) host further corroborated the conclusion that CD74 and DM are both required for the expression of the DQβ0603:DQα0103 reactive epitope. The strong reactivity to the recombinant DQβ0603:DQα0103 proteins from the HLA class I-expressing T2DM and LCL3023 hosts and negative reactivity to the HLA class I-negative LCLKO host supported the observation we showed in Fig. 1a that HLA class I peptides were participating in forming the epitope recognized by S1-S5.

The transfectants were further tested by flow cytometry for the surface expression and assembly of the DQβ0603:DQα0103 protein with the S1 serum to confirm the reactivity observed on the SAB assay. The expression of the HLA class I proteins was at a similar level between the T2 and T2DM transfectants, however, the T2DM transfectant has a slightly higher expression of HLA DQ than the T2 transfectant. The positive staining of S1 with the T2DM transfectant,

but negative staining with T2 transfectant supports the SAB results suggesting that HLA-DM was essential for the assembly of the epitope recognized by S1 (Fig. 2c). The LCL3023 transfectant had a higher level of expression for HLA class I and a similar level of HLA DQ expression compared to LCLKO transfectant. However, S1 had stronger staining on LCL3023 transfectant than on the LCLKO transfectant, even though they have a similar level of HLA DQ staining, confirmed the participation of HLA class I derived peptides in the S1 reactivity (Fig. 2d).

The negative (or very weak) reactivity of S6 on recombinant DQβ0603:DQα0103 protein derived from HLA class I-expressing T2DM and LCL3023, suggests that the peptides derived from the HLA class I alleles expressed in T2DM or LCL3023 could not provide the epitope recognized by S6.

## Expression of HLA class I gene in DQB1*06:03-DQA1*01:03 transfectants rescued the DQβ0603:DQα0103 antibody reactive epitope

Since the DQβ0603:DQα0103 protein isolated from 9058 B cell line was reactive with S6, the HLA A*02:01 or C*16:01 alleles from 9058 cell line (Table 1) were introduced into the LCLKO transfectant expressing DQβ0603:DQα0103 antigen to determine whether peptides derived from C*16:01 or A*02:01 could restore the S6 or S1 epitope reactivity, respectively. The recombinant DQβ0603:DQα0103 proteins were extracted from the co-

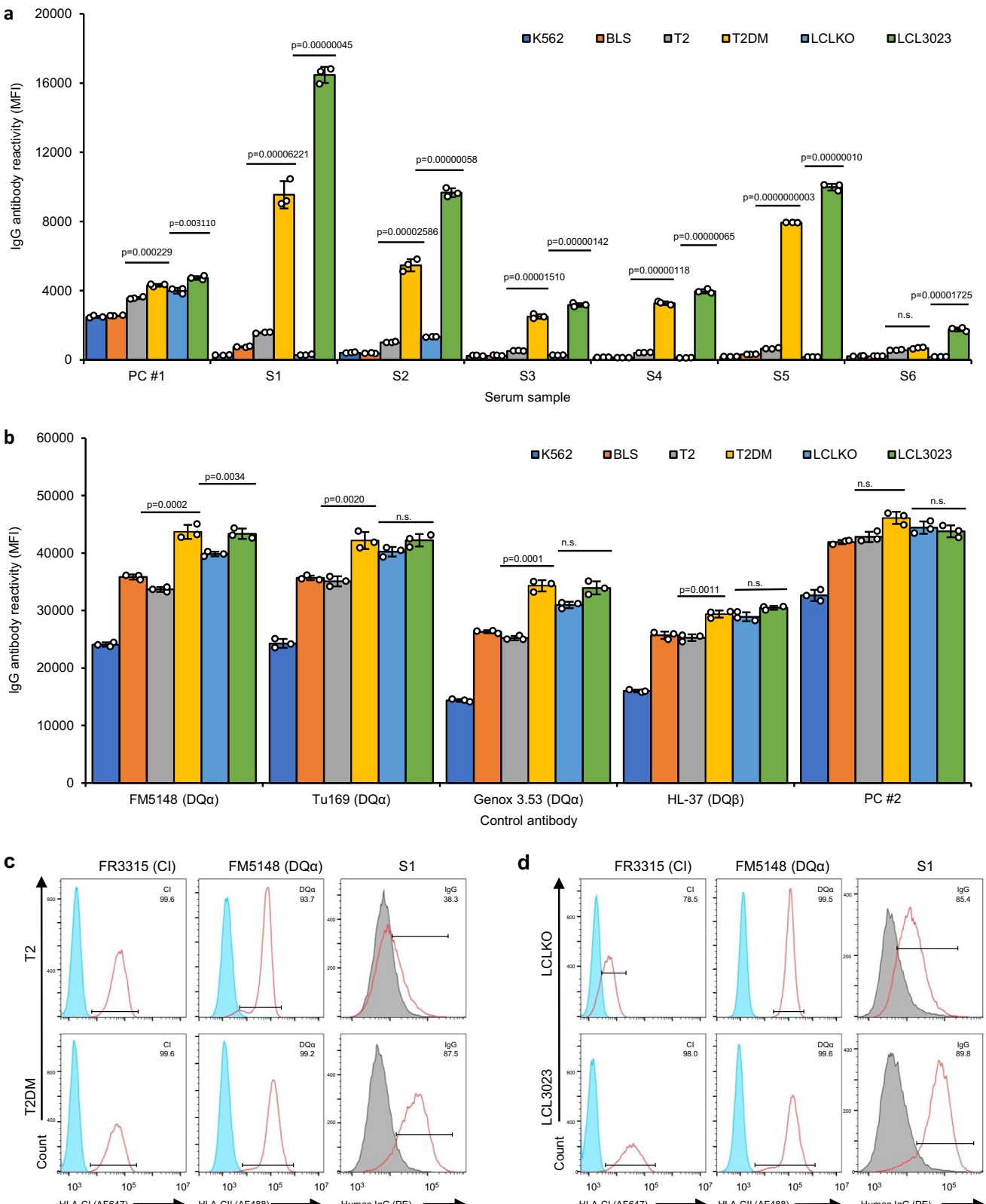

transfectants and their antibody reactivity compared to the proteins purified from the original LCLKO transfectants. The expression of A*02:01 rescued the serum reactivity of S1 (Fig. 3a), but expression of C*16:01 did not (Fig. 3c). Conversely, incorporation of C*16:01 rescued the serum reactivity of S6 (Fig. 3c), but the incorporation of A*02:01 did not (Fig. 3a), supporting the observation that S6 negative reactivity in Fig. 2 was due to the missing specific

HLA class I peptide in T2DM and LCL3023 hosts and was not caused by other possible mechanisms. Expression of the additional class I gene in the transfectants didn't affect the reactivity of proteins to the four monoclonal antibodies and PC#2 that recognize the DQα and DQβ chains (Figs. 3b and 3d). These data strongly implicate the involvement of a specific HLA class I polymorphic sequence in the DQβ0603:DQα0103 antibody reactivity.

**Fig. 2 | Reactions of DQβ0603:DQα0103 specific sera to the recombinant DQβ0603:DQα0103 protein from different host cell lines.** The antibody reactivity of DQβ0603:DQα0103 protein extracted from transfectants of six different host cell lines, K562 (dark blue), BLS (orange), T2 (gray), T2DM (yellow), LCLKO (light blue) and LCL3023 (green), were measured as mean fluorescence intensity (MFI) using the LABScreen SAB assay protocol. Data are presented as mean values +/- SD. Bars indicate the mean, whiskers indicate the standard deviation, and white dots with black circle represent the individual data points from triplicate tests (*n* = 3). Statistical significance was determined using one-way ANOVA followed by Bonferroni post hoc test. The complete statistical analysis results are provided in Source data file. Only results that compare the related hosts (T2 versus T2DM and LCLKO versus LCL3023) are shown. n.s. represents no significant difference. **a** With six DQβ0603:DQα0103 specific sera and a weak positive control serum, PC #1. **b** With four monoclonal antibodies. FM5148 and Tu169 are specific for a monomorphic epitope on DQα, Genox 3.53 is specific for DQα01 and HL-37 is specific for DQβ01,03 and a strong positive control serum, PC #2. **c** Staining patterns of FR3315 (anti-HLA class I), FM5148 (anti-HLA DQα) and S1 on T2 or T2DM were shown in red line; the isotype control was shown as filled blue histogram. **d** Staining patterns of FR3315 (anti-HLA class I), FM5148 (anti-HLA DQα) and S1 on LCLKO or LCL3023 were shown in red line. And HLA negative serum staining was shown as filled gray histogram.

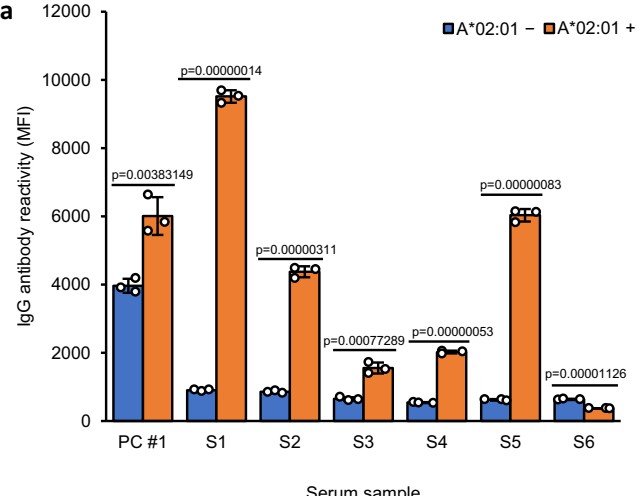

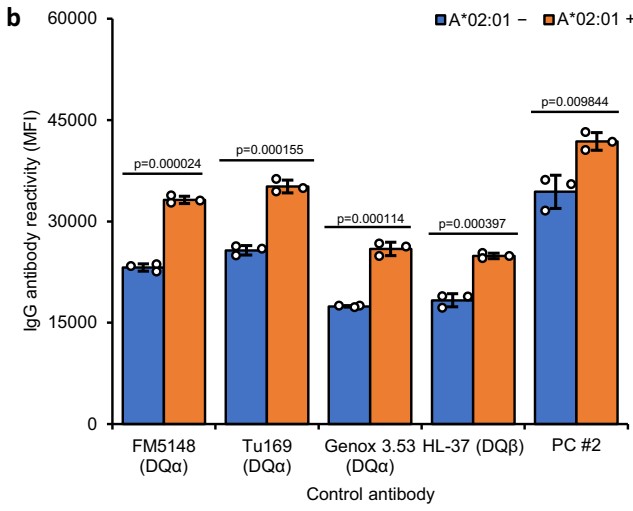

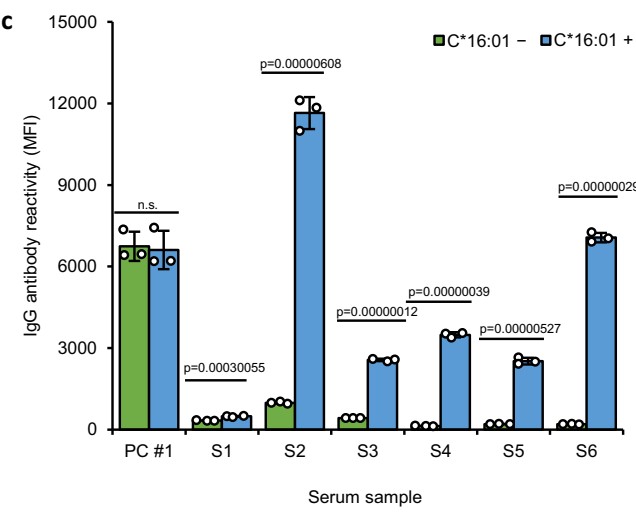

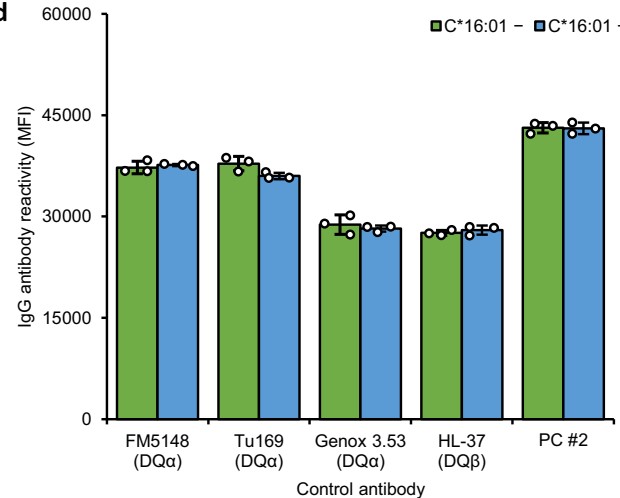

**Fig. 3 | Reactions of DQβ0603:DQα0103 specific sera to recombinant DQβ0603:DQα0103 protein purified from LCLKO with A\*02:01 or C\*16:01 gene.** The antibody reactivity of DQβ0603:DQα0103 protein extracted from co-transfectants of DQB1\*06:03, DQA1\*01:03, A\*02:01 (orange) or DQB1\*06:03, DQA1\*01:03, C\*16:01 (light blue) of LCLKO host cell line were compared to the DQβ0603:DQα0103 protein extracted from DQB1\*06:03, DQA1\*01:03 transfected LCLKO (dark blue or green), and were measured as mean fluorescence intensity (MFI) using the LABScreen SAB assay protocol. Data are presented as mean values +/- SD. Bars indicate the mean, whiskers indicate the standard deviation, and white dots with black circle represent the individual data points from triplicate tests (*n* = 3). Statistical significance was defined by Student's t-test (*p* < 0.05). n.s. represents no significant difference. The complete statistical analysis results are provided in Source data file. **a, c** With six DQβ0603:DQα0103 specific sera and a weak positive control serum, PC #1. **b, d** With four monoclonal antibodies, FM5148 and Tu169 specific for a monomorphic epitope on DQα, Genox 3.53 specific for DQα01 and HL-37 specific for DQβ01,03, and a strong positive control serum, PC #2.

**Table 2 | HLA class I peptides identified from the DQβ0603:DQα0103 protein peptidome analysis**

| 9060 | Position | | | 9062 | Position | | |
|---|---|---|---|---|---|---|---|
| Allele | start | end | Amino Acid Sequence | Allele | Start | end | Amino Acid Sequence |
| A*01:01 | 119 | 136 | DGKDYIALNEDLRSWTAA | A*02:01 | 129 | 144 | DLRSWTAADMAAQTTK |
| | 129 | 145 | DLRSWTAADMAAQITKR | | 129 | 145 | DLRSWTAADMAAQTTKH |
| | 128 | 144 | EDLRSWTAADMAAQITK | | 128 | 144 | EDLRSWTAADMAAQTTK |
| | 128 | 145 | EDLRSWTAADMAAQITKR | | 128 | 145 | EDLRSWTAADMAAQTTKH |
| | | | | | 128 | 146 | EDLRSWTAADMAAQTTKHK |
| B*15:01 | 128 | 144 | EDLSSWTAADTAAQITQ | B*38:01 | 128 | 148 | EDLSSWTAADTAAQITQRKWE |
| | | | | | 126 | 145 | LNEDLSSWTAADTAAQITQR |
| C*03:03 | 119 | 141 | DGKDYIALNEDLRSWTAADTAAQ | C*:12:03 | 128 | 141 | EDLRSWTAADTAAQ |
| | 128 | 141 | EDLRSWTAADTAAQ | | 128 | 144 | EDLRSWTAADTAAQITQ |
| | 128 | 143 | EDLRSWTAADTAAQIT | | 128 | 145 | EDLRSWTAADTAAQITQR |
| | 128 | 144 | EDLRSWTAADTAAQITQ | | 128 | 148 | EDLRSWTAADTAAQITQRKWE |
| | 128 | 145 | EDLRSWTAADTAAQITQR | | 126 | 141 | LNEDLRSWTAADTAAQ |
| | 128 | 148 | EDLRSWTAADTAAQITQRKWE | | 126 | 144 | LNEDLRSWTAADTAAQITQ |
| | 126 | 141 | LNEDLRSWTAADTAAQ | | | | |
| | 127 | 141 | NEDLRSWTAADTAAQ | | | | |
| E | 128 | 141 | EDLRSWTAVDTAAQ | E | 128 | 141 | EDLRSWTAVDTAAQ |
| | 128 | 144 | EDLRSWTAVDTAAQISE | | 129 | 144 | DLRSWTAVDTAAQISE |
| | 126 | 141 | LNEDLRSWTAVDTAAQ | | | | |
| | 129 | 141 | DLRSWTAVDTAAQ | | | | |
| | 130 | 141 | LRSWTAVDTAAQ | | | | |
| F | 128 | 141 | EDLRSWTAADTVAQ | F | 128 | 141 | EDLRSWTAVDTAAQ |
| | | | | | 129 | 144 | DLRSWTAVDTAAQISE |
| | | | | DRA | 126 | 143 | KPVTTGVSETVFLPREDH |

## Polymorphic HLA class I peptides were identified to associate with the DQβ0603:DQα0103 proteins

In order to identify the specific HLA class I peptides that were presumably bound to the groove of DQβ0603:DQα0103 proteins and recognized by S1, proteins isolated from two B cell lines, 9062 which reacted strongly to S1 and 9060, which was weakly reactive to S1, were selected for HLA peptidome analysis.

Multiple peptides located between amino acid position 119-148 of the mature HLA class I heavy chain were identified by mass spectrometry analysis (Table 2). Interestingly, only peptides from this region were identified from the peptidome analysis regardless of whether the peptides were derived from HLA-A, -B, -C, -E or -F. Peptides that contain a polymorphic sequence at amino acid position 142 for HLA A*02:01 were present in the antigen prepared from 9062 and for A*01:01 in antigen prepared from 9060 suggesting the polymorphic A*02:01 peptide might be involved in the S1 antibody epitope.

## DQβ0603:DQα0103 antibodies recognize different amino acid sequences of same but polymorphic class I region

To confirm the expected peptide fragments of the class I proteins that participated in the DQβ0603:DQα0103 reactive epitope, one peptide with the same length between 9060 and 9062 from each class I gene in the peptidome analysis was synthesized and used to verify the ability to restore the antibody reactivity after loading it onto HLA class I negative recombinant DQβ0603:DQα0103 proteins in Fig. 2a. Additionally, peptides with the sequences from the same location of other common alleles that were not presented in the peptidome analysis were also synthesized and tested. Biotin was incorporated at the N-terminal of the peptides to measure the efficiency of the peptide loading process (Table 3). Peptides were loaded onto the DQβ0603:DQα0103 protein from the bead panel in Fig. 2a and their peptide binding efficiencies were measured by the reactivity of R-phycoerythrin conjugated streptavidin (SAPE) with biotin at the

**Table 3 | Synthetic peptide sequence and the matching serological groups**

| ID | Sequence | HLA sero-group |
|---|---|---|
| A1* | Bio-EDLRSWTAADMAAQITKR | A1, A3, A11, A24, A36, A80 |
| A2* | Bio-EDLRSWTAADMAAQTTKH | A2, A68, A69 |
| A3 | Bio-EDLRSWTAADMAAQITQR | A23, A25, A26, A29, A30, A31, A32, A33, A34, A43, A66, A74 |
| B1* | Bio-EDLSSWTAADTAAQITQRKWE | B14, B15, B18, B27, B35, B37, B38, B39, B44, B45, B46, B47, B49, B50, B51, B52, B53, B54, B55, B56, B57, B58, B59, B67, B78, B82 |
| B2 | Bio-EDLSSWTAADTAAQITQLKWE | B13 |
| B3 | Bio-EDLRSWTAADTAAQISQRKLE | B60, B48, B81, C17 |
| C1* | Bio-EDLRSWTAADTAAQITQRKWE | B7, B8, B61, B41, B42, C1, C2, C3, C4, C6, C7, C0801, C12, C14, C15, C16, C18 |
| C2 | Bio-EDLRSWTAADKAAQITQRKWE | C5, C0802 |
| E1* | Bio-EDLRSWTAVDTAAQISE | E |
| F1* | Bio-EDLRSWTAADTVAQITQ | F |
| DRA1* | Bio-KPVTTGVSETVFLPREDH | DRA |

Peptides with * were derived from the peptidome analysis in Table 2.

N-terminal of the peptides. The DQβ0603:DQα0103 protein from BLS cell line showed the highest SAPE signals with all the peptides (Supplementary Table 1) and its interaction with the DQβ0603:DQα0103 antibodies was examined. All six sera reacted to at least one peptide loaded protein (Fig. 4a–f). SAPE reacted to all HLA peptides with similar intensity confirming the binding of HLA peptides by sera S1-S6 were peptide-specific (Fig. 4g). Serum S1 reacted primarily with A2 peptide from A*02:01 and weakly with peptides from other HLA alleles including A1 peptide which sequence was present in HLA A locus alleles

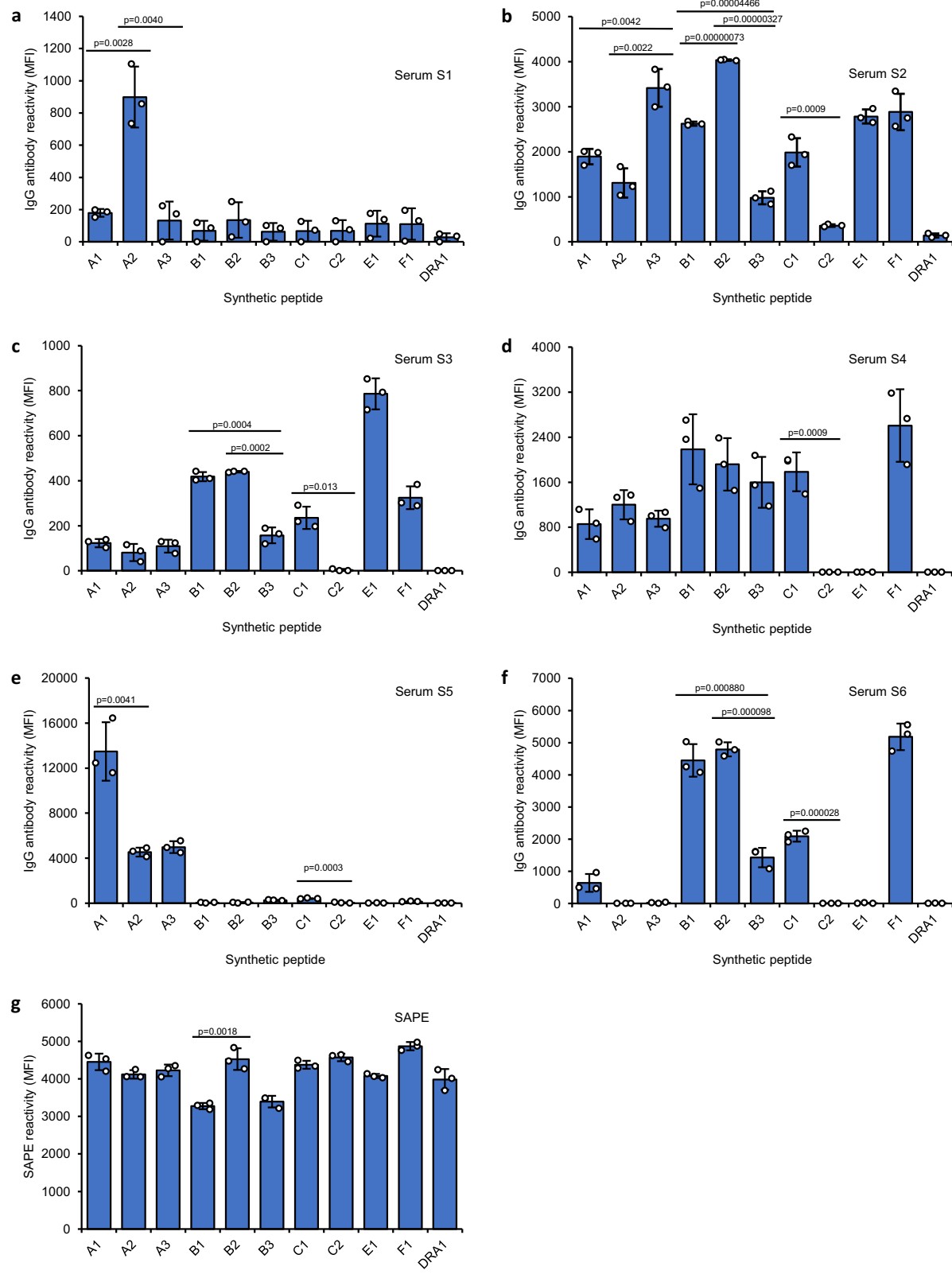

from weak S1 reactive B cell lines, 9060, 9065 and 9105 (Fig. 4a). There is only one amino acid difference at amino acid position 142 between A2 and A1 peptide that determined the specificity of the antibody in Serum S1. Serum S6 reacted to a core sequence "SWTAADT" shared by most of the HLA B and some C alleles and HLA-F and was completely negative with the three peptides in A locus (A1-A3) confirming the

negative reactivity with proteins isolated from the T2DM and LCL3023 transfectants seen previously (Fig. 4f).

Serum S5 reacted to all three A locus sequences suggesting that the recognition site might be centered on the M (Methionine) at amino acid position 138 and the nearby sequence but does not extend past Q (Glutamine) at amino acid position 141 (Fig. 4e). Assuming all these

**Fig. 4 | Avidity of DQβ0603:DQα0103 sera to DQβ0603:DQα0103 protein loaded with various HLA class I heavy chain peptides.** DQβ0603:DQα0103 protein from BLS transfectant was loaded with various synthetic HLA class I heavy chain peptides and then used to evaluate the serum reactivity of S1-S6 (**a**–**f**). The antibody reactivity was measured as mean fluorescence intensity (MFI) using the LABScreen SAB assay protocol. Data are presented as mean values +/- SD. Bars indicate the mean, whiskers indicate the standard deviation, and white dots with black circle represent the individual data points from triplicate tests ($n$ = 3). The antibody reactivity to each peptide loaded antigen was calculated by subtracting the reactivity of the same antigen incubated without the peptide. The values were adjusted to 0 when lower than 0. Successful peptide loading for all peptides was demonstrated by the strong SAPE interaction with biotin at the N-terminal of the peptides (**g**). Statistical significance was determined using one-way ANOVA followed by Bonferroni post hoc test. Only Bonferroni-adjusted $p$ values of <0.004545 are shown. The complete statistical analysis results are provided in Source data file. Only results that compare the peptides in the same locus are shown.

## Table 4 | Amino acid sequence differences of HLA-DQβ and -DQα between the donor and the recipient in S22

| Amino acid position | 30 | 57 | 70 | 86 | 87 | 130 |
|---|---|---|---|---|---|---|
| DQB1*06:03(donor) | H | D | G | A | F | R |
| DQB1*06:09(recipient) | Y | V | R | G | Y | Q |
| DQB1*04:02(shared) | Y | D | E | E | L | R |
| Amino acid position | 25 | 41 | 129 | 130 | 207 | |
| DQA1*01:03(donor) | F | K | H | A | V | |
| DQA1*01:02(recipient) | Y | R | Q | S | M | |
| DQA1*04:01(shared) | Y | R | H | S | V | |

peptides were binding to the groove of DQβ0603:DQα0103 through the same anchor positions, it suggested that the amino acid sequences of DQβ0603:DQα0103 participating in the antibody epitopes were not the same for all the DQβ0603:DQα0103 antibodies. Serum S4 reacted to all class I peptides except C2 and E with the likely recognized sequence of 136ADT/M138 (Fig. 4d). The positive charge of 138 K on C2 was most likely the reason for the negative reaction of S4. Serum S2 reacted to most of the peptides with various degrees of intensity (Fig. 4b). Serum S3 has similar but weaker reactivity than S2 (Fig. 4c).

The different reactivity observed for peptides from amino acid position 128-145 of the polymorphic region on class I heavy chain loaded in DQβ0603:DQα0103 protein supports the conclusion that DQβ0603:DQα0103 antibodies can differentiate peptides with polymorphic sequences from different loci and alleles.

### DQβ0603:DQα0103 antibodies from different individuals demonstrated varying avidity profiles against the panel of class I peptides

Since the six DQβ0603:DQα0103 sera studied differed in their reactivity profiles when binding to the DQβ0603:DQα0103 protein loaded with different peptides, we decided to evaluate an additional fifteen DQβ0603:DQα0103 reactive sera for their ability to react to the DQβ0603:DQα0103 protein loaded with the same panel of peptides presented in Table 3. Each of the fifteen sera had its own unique reactivity profile for the DQβ0603:DQα0103 protein loaded with different peptides (Fig. 5a–o). These reactivities differed from the profiles of the initial six sera in Fig. 4. Only three of the cases (5d, 5 g and 5 m) could be explained by simple amino acid mismatches among the peptides. The reactivity profiles for the remaining cases were not obvious from the peptide sequence alignment and could be the result of either multiple independent antibodies or the accessibility of the mismatching amino acids of the bound peptides. The observation of unique reactivity profiles for every serum in this study demonstrated that the antibody avidity generated by different individuals to the same DQβ0603:DQα0103 protein could be a function of the bound peptide repertoire.

Since peptides A3, B2, B3 and C2 were synthesized based on their identical location and homology to the HLA class I proteins with the peptides detected in the peptidome analysis, identification of sera that reacted to DQβ0603:DQα0103 protein loaded with A3 peptide (Fig. 5m, n), with B2 peptide (Fig. 5a), with B3 peptide (Fig. 5j and k) or with C2 peptide (Fig. 5d) suggests this approach can be applied to investigate peptide polymorphism in antibody interaction with other class II proteins.

### De-novo HLA class I peptide-dependent DQβ0603:DQα0103 DSA identified in a post-transplant patient

A serum sample (S22) from a kidney transplant patient who received a DQβ0603:DQα0103 mismatched kidney (Table 4) was identified to have de-novo DSA against the DQβ0603:DQα0103 antigen six years post-transplant (Supplementary Fig. 1a) and it was the only DQ specificity detected (Supplementary Fig. 1b). This patient was biopsied in the month of the appearance of the de novo DSA due to an elevated creatinine (previously stable) and the biopsy histological analysis showed suspicion for AMR. The patient was treated with Solu-Medrol, plasma exchange and intravenous immunoglobulin (IVIg). There were no known immunizing events, and the patient was compliant, however, there was chronic (moderate) cyclosporin underdosing since the patient was intolerant to other immunosuppressive regimens. There was no improvement in renal function after AMR treatment and current status is unknown.

The antibody reactivity in S22 was further characterized using the same recombinant DQβ0603:DQα0103 antigen panel in Fig. 2 (Fig. 6a). The reaction pattern of S22 was similar to S1-S5 which was only positive to the antigens purified from the CD74 +, DM+ and HLA class I expressing host cell lines. Also, similar to S1-S5, introduction of additional HLA-A*02:01 or C*16:01 gene into LCLKO transfectant rescued the reactivity to the DQβ0603:DQα0103 protein (Fig. 6b). These results suggest that the detected HLA class I peptide-associated DQβ0603:DQα0103 reactive antibody is clinically relevant in the setting of AMR.

## Discussion

Many new HLA alleles have been identified in recent years due to the advances in molecular typing technology, but the serological reactivity of a majority of these alleles has not been determined. Various algorithms have been developed that aim to address this problem, and improve the accuracy of the VXM by defining antibody reactive epitopes based on HLA protein sequence alignment and 3D structure modeling[17,18]. Antibody reactivities that could not be assigned to epitopes by such algorithms were frequently questioned for their validity[19]. In this report, we present evidence that a different type of antibody epitope does exist. This unique epitope is based in part on the peptides bound to the Major Histocompatibility Complex (MHC) binding groove of HLA class II antigens. Our findings are based on the extensive investigation of twenty-one DQβ0603:DQα0103 reactive sera from patients that are not carrying the DQB1*06:03/DQA1*01:03 genotype (Supplementary Table 2). The peptides bound to the DQβ0603:DQα0103 antigen in our study were found to be mainly derived from a polymorphic region of HLA class I proteins, and these twenty-one sera recognized different portions of the intra-locus or inter-locus polymorphic peptides in association with the DQβ0603:DQα0103 dimer to collectively form the antibody reactive epitope. The participation of the polymorphic residue can either be direct through interaction with the exposed sidechain or indirect through peptide interaction with the base of the binding groove

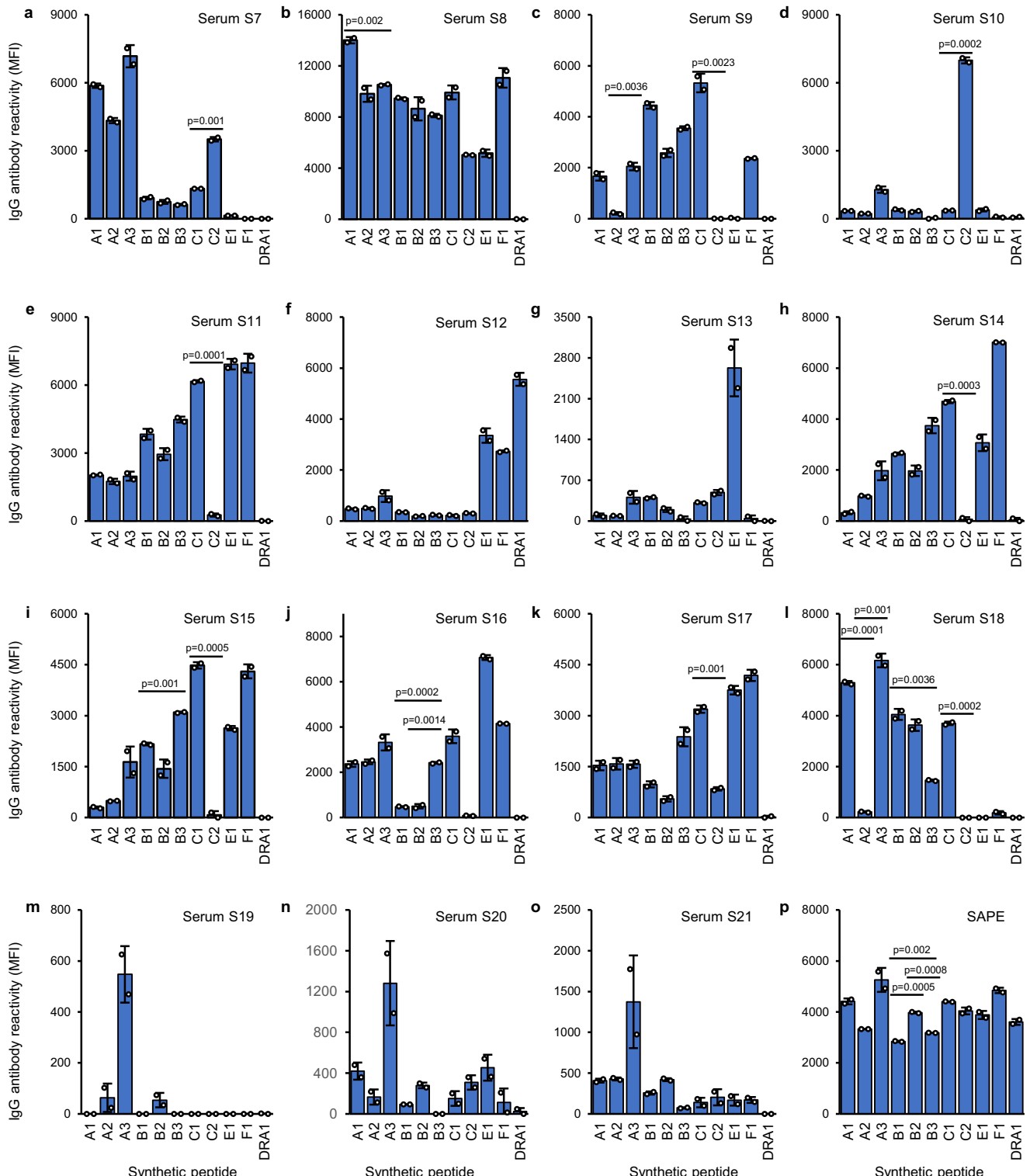

**Fig. 5 | Avidity of 15 DQβ0603:DQα0103 sera to DQβ0603:DQα0103 protein loaded with various HLA class I heavy chain peptides.** DQβ0603:DQα0103 protein from BLS transfectant was loaded with various HLA class I heavy chain peptides and then used to evaluate the serum reactivity of S7-S21 (**a**–**o**). The antibody reactivity to each peptide loaded antigen was calculated by subtracting the reactivity of the same antigen incubated without the peptide. The values were adjusted to 0 when lower than 0. Successful peptide loading for all peptides was demonstrated by the strong SAPE interaction with biotin at the N-terminal of the peptides (**p**) Data are presented as mean values +/- SD. Bars indicate the mean, whiskers indicate the standard deviation, and white dots with black circle represent the individual data points from duplicate tests (*n* = 2). Statistical significance was determined using one-way ANOVA followed by Bonferroni post hoc test. Only Bonferroni-adjusted *p* values of <0.004545 are shown. The complete statistical analysis results are provided in Source data file. Only results that compare the peptides in the same locus are shown.

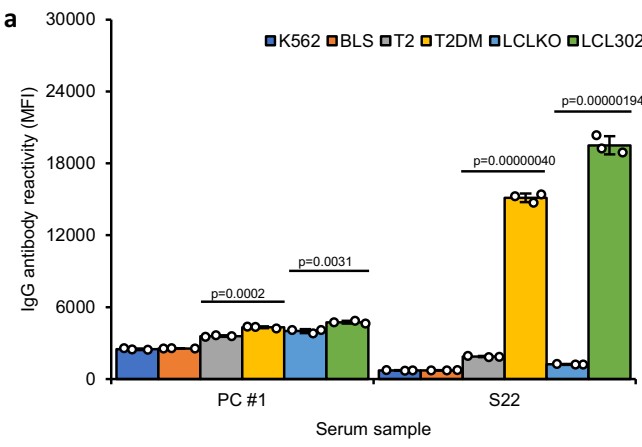

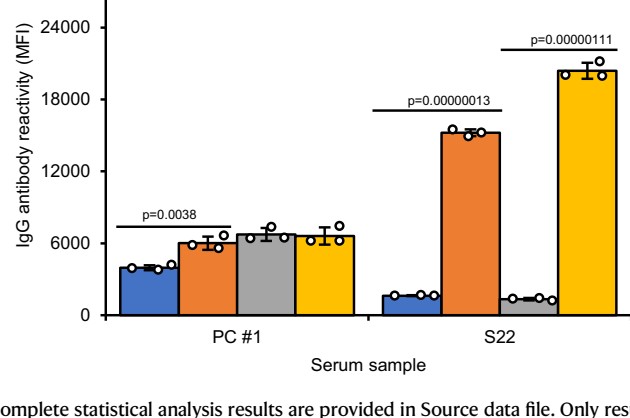

**Fig. 6 | Characterization of the de novo DQβ0603:DQα0103 antibody in Serum S22.** S22 and PC#1 were tested on two different DQβ0603:DQα0103 antigen panels. The antibody reactivity was measured as mean fluorescence intensity (MFI) using the LABScreen SAB assay protocol. Data are presented as mean values +/- SD. Bars indicate the mean, whiskers indicate the standard deviation and white dots with black circle represent the individual data points from triplicate tests (*n* = 3). **a** The same bead panel of the DQβ0603:DQα0103 proteins from transfectants of six different host cell lines as in Fig. 2, K562 (dark blue), BLS (orange), T2 (gray), T2DM (yellow), LCLKO (light blue) and LCL3023 (green). Statistical significance was determined using one-way ANOVA followed by Bonferroni post hoc test. The

complete statistical analysis results are provided in Source data file. Only results that compare the related hosts (T2 versus T2DM and LCLKO versus LCL3023) are shown. **b** With the same bead panel of HLA A*02:01 or C*16:01 co-transfected DQβ0603:DQα0103 proteins as in Fig. 3. The antibody reactivity of DQβ0603:DQα0103 protein extracted from co-transfectants of DQB1*06:03, DQA1*01:03, A*02:01 (orange) or DQB1*06:03, DQA1*01:03, C*16:01 (light blue) of LCLKO host cell line were compared to the DQβ0603:DQα0103 protein extracted from DQB1*06:03, DQA1*01:03 transfected LCLKO (dark blue or green). Statistical significance was defined by Student's t-test (*p* < 0.05). The complete statistical analysis results are provided in Source data file.

altering the structure of the protein and thus influence antibody binding[20].

Among the twenty-one patients who possessed the DQβ0603:DQα0103 reactivity, twelve had been transplanted with preformed DQβ0603:DQα0103 non-DSA antibody and nine are still awaiting transplant. Most of the twenty-one patients had no documented sensitization event when the antibody was first detected. Interestingly, all twenty-one sera identified reacted to epitopes involving HLA class I peptides, even though the DQβ0603:DQα0103 protein is capable of binding many other peptides (ProteomeXchange Consortium identifier PXD043999). Surprisingly, in the patient that produced DSA toward the DQβ0603:DQα0103 protein (S22) post-transplant the detected antibody appeared to react with the same HLA class I peptide-dependent epitope as the other twenty-one sera even though there were significant differences in amino acid sequences between the mismatched DQα and DQβ chains between the donor and recipient (Table 4). This observation suggests that the identified HLA class I peptide-specific DQβ0603:DQα0103 epitopes are immunogenic and clinically relevant.

Our report demonstrate that human HLA class II antibody reactivity can be influenced by polymorphic peptides, derived from the HLA class I proteins, bound to HLA class II peptide binding groove. We anticipate that other similar types of antibodies might be influenced by peptides derived from other proteins that are also polymorphic in nature.

Our discovery has multiple implications. For transplantation, it is critical to correctly identify antibody-binding epitopes based on SAB data and HLA typing in order to facilitate an accurate VXM. Since current SAB technologies are dependent on recombinant HLA proteins generated in the context of a limited HLA class I background, these technologies might underestimate peptide dependence of HLA antibody reactivity in a given patient that has been immunized in the setting of a different HLA class I background. Current molecular based analysis of graft biopsies suggests that in up to 50% of patients with AMR and documented evidence of graft damaged as assessed by elevated donor derived cell-free DNA, do not have a detectable DSA based on current SAB assays[21,22]. It has been suggested that non-HLA antibodies are responsible for the AMR in these cases but perhaps, a

portion of these patients may have peptide-dependent HLA antibodies that cannot be detected with the current SAB assays. For the field of basic immunology our findings could be provocatively interpreted to suggest that B cells, through their B cell receptor (BCR), are able to recognize peptides bound to MHC class II molecules and through this recognition generate antibodies directed at a specific MHC peptide combination. If future studies are able to confirm our findings this could affect the way we view B cell function and adaptive immunity.

The cell-based flow cytometric crossmatch is often used to confirm the reactivity of antibodies detected with the SAB assay. However, since the expression of DQ molecules on the cell surface is significantly lower than that of DR molecules[12], the detection of such antibodies may be challenging. Moreover, if only a fraction of those DQ molecules have the correct peptide that is necessary for reactivity, this may further add to the frequency of discordant results between cell-based and bead-based assays. Indeed, several previous studies have shown an increased risk of AMR in the setting of SAB DSA positive but flow cytometric crossmatch negative transplants[23–25]. Pre-transplant or de novo DSA directed towards HLA-DQ have also been associated with particularly poor outcomes in kidney transplantation, which further emphasizes the potential clinical relevance of our findings[10,26,27].

The peptides that can bind to DQβ0603:DQα0103 protein are restricted by their anchor residues that fit into the MHC binding groove of the class II antigens[28] and those restricted peptides in turn contribute to the antibody specificity. Changes in the amino acid sequences of the α or β chain that can affect the peptide-binding groove could result in a different peptide repertoire thereby altering the binding specificity of an antibody. This could occur even if two alleles have the same amino acid sequence in the surrounding alpha helix protein structure that is used by epitope prediction algorithms.

Further studies on the role of peptides in defining antibody reactivity are needed to understand their frequency and their role in defining the antibody epitope. In summary, we present evidence suggesting that HLA antibody reactivity with the DQβ0603:DQα0103 protein detected in patient sera both pre- and post-transplant is in part dependent on the presented peptide. Our findings have potentially important implications for the field of transplant immunology and for our understanding of adaptive immunity.

## Methods

### Identification of DQβ0603:DQα0103 specific sera

Sera from kidney patients on the wait list of each transplant center were routinely screened on LABScreen class II single antigen panel (One Lambda Inc., West Hill, CA) and those that reacted only to the DQβ0603:DQα0103 antigen beads among all the DQ antigen beads with the signal greater than 3000 mean fluorescence intensity (MFI) were identified for further characterization. The use of human sera was approved and written consent from patients were waived by the Methodist Specialty and Transplant IRB Board of Directors, Emory University IRB, Cantonal Ethics Committee in Zurich and Comité de Protection des Personnes de Paris Ile de France 4.

### Cell lines and antibodies

EBV transformed B cell lines, 9058, 9060, 9062, 9065, and 9105 from the 10th International Histocompatibility Workshop and Conference (IHWC) were the gift from Dr. Jennifer Ng. Epstein–Barr virus (EBV) transformed B cell lines, LCL3023 and bare lymphocyte syndrome (BLS) B cell line were gifts from Drs. William Burlingham and Roland Martin, respectively. K562 and T2 cell lines were purchased from ATCC (Manassas, VA). T2DM, a T2 cell line transfected with HLA-DM (DMA*01:01 and DMB*01:01) cDNA and LCLKO were kindly provided by One Lambda. T2 was derived from LCL 721.174[29] which has deletions from DRA to DPB1 in both copies of the chromosomes. LCL3023[15] is from the LCL 721.174 transfected with both HLA-DM and HLA-DRA genomic DNA. LCLKO was derived from LCL721.221[30]. All BLS, K562, LCL3023, LCLKO, T2 and T2DM cell lines have no detectable expression of HLA class II on the cell surface. HLA monoclonal antibody FM5148 (DQα) and FR3315 (class I) were gifts from One Lambda. DQ1 antibody Genox 3.53 (Catalog # MA1-46297), DQ1,3 antibody HL-37 (cat #MA1-19143) and mouse IgG isotype control MOPC-21 (cat # MA1-10407) were purchased from Thermo Fisher Scientific (Waltham, MA). Tu169 (cat #361502) was purchased from BioLegend (San Diego, CA). For all monoclonal antibodies, one microgram (µg) was used for each reaction in a 25 microliter (µl) reaction volume. R-Phycoerythrin labeled Goat Anti-Mouse IgG (cat #115-116-146) and R-Phycoerythrin labeled Goat Anti-Human IgG (cat #115-116-146) secondary antibody were purchased from Jackson ImmunoResearch Laboratories (West Grove, Pennsylvania) and were reconstituted with 1 milliliter (ml) water. The reconstituted antibody was diluted 1:100 in the LABScreen wash buffer and 100 µl of the diluted antibody was used per reaction.

### Transfection and cell culture

The DQA1*01:03 and DQB1*06:03 cDNA sequences as well as HLA class I sequences were synthesized according to the published sequences in the IPD-IMGT/HLA Database (https://www.ebi.ac.uk/ipd/imgt/hla/). DQA1*01:03 and DQB1*06:03 alleles were cloned into the pEF6 mammalian expression vector (Thermo Fisher Scientific, Waltham, MA, cat # V963-20). HLA class I gene expression vectors were gifts from One Lambda. Plasmids were co-transfected into human cell lines as mentioned in the text, using a Neon Transfection System (Thermo Fisher Scientific, Waltham, MA, cat # MPK5000) according to the manufacturer's protocol. Cells were pulsed twice with 1,100 volts for a duration of 30 ms (millisecond). Stable cells were established by RPMI-1640 media (Thermo Fisher Scientific, Waltham, MA, cat # 11875085) with 10% fetal bovine serum (FBS) (Thermo Fisher Scientific, Waltham, MA, cat # A5670801) and with corresponding selection antibiotics according to the plasmids used (Thermo Fisher Scientific, Waltham, MA, cat # A1113902).

### HLA antibody assay

HLA class II antigens were purified by cell lysis and affinity purification from cell pellets using HLA-DQ specific antibody FM5148 and attached to Luminex beads[4]. Solid-phase bead assay was performed as per the LABScreen antibody detection protocol and analyzed by a LABScan3D flow analyzer (One Lambda, West Hills, CA).

### Flow cytometry analysis and cell sorting for transfectants

Monoclonal antibodies were first labeled with Fluorochrome using Alexa Fluor™ Antibody Labeling Kits (ThermoFisher Scientific, Waltham, MA, cat # A20181 or A20186) and then 1 µg was used to label $10^6$ cells in 100 µl 1X PBS, 0.1% glucose and 2% FBS (cell wash buffer) for 30 min at 4 °C. Following labeling, cells were washed twice with cell wash buffer and resuspended in 100 µl cell wash buffer. Flow cytometric analyses were performed in a Quanteon Flow Cytometer (Agilent, Santa Clara, CA). Flow gating strategies were illustrated in Supplementary Fig. 2. Data was analyzed with FlowJo Software v10.8. Stable transfected cells stained with FM5148-Alexa Fluor 488 were sorted on a MA900 cell sorter (Sony Biotechnology, San Jose, CA). Sorting strategies were demonstrated in Supplementary Fig. 3a and b. Cells with high expression of HLA-DQα (FM5148-Alexa Fluor 488) were sorted and plated in RPMI-1640 + 10% FBS with the corresponding selection antibiotics.

### Flow cytometry analysis with human sera

Approximately $10^6$ cells from the stable transfectants were first treat with LIVE/DEAD™ Fixable Near-IR Dead Cell Stain Kit (Thermo Fisher Scientific, cat # L34975) according to manufactural procedure. In brief, 1 µl dye is mixed with $10^6$ cells in 1 ml 1X PBS for 30 min at 4 °C and then blocked by cell wash buffer. The treated cells were resuspended in 100 µl of cell wash buffer before mixing with 50 µl of test serum, and incubated for 20 minutes at 4 °C. After incubation, cells were washed with cell wash buffer and then incubated with 1 µg of biotinylated F(ab')2-Goat anti-Human IgG Fc-specific antibody (Invitrogen, cat # A24480) in 100 µl cell wash buffer for 20 minutes. Samples were washed and incubated with 1 µl SAPE (Jackson ImmunoResearch Laboratories, West Grove, PA, cat # 016-110-084) in 100 µl cell wash buffer for 20 minutes at 4 °C. After washing, samples were acquired on a Quanteon Flow Cytometer (Agilent Technologies, Santa Clara, CA). Data was analyzed with FlowJo Software (https://www.flowjo.com/). To check the HLA expression, cells were labeled with 1 µg of fluorochrome-conjugated antibodies, Alexa Fluor® 488 conjugated FR3315 (HLA class I) or Alexa Fluor® 647 conjugated FM5148 (HLA-DQα) in 100 µl cell wash buffer. One µg of Alexa Fluor® 488 or 647 conjugated mouse IgG1 isotype controls, clone MOPC-21, in 100 µl cell wash buffer were used to establish the baseline for FM5148 or FR3315, respectively. Forty thousand events (ungated) were acquired.

### HLA peptidome analysis

Bound HLA antigens from the affinity matrix with monoclonal antibody FM5148 were eluted and processed without the presence of detergents. Peptide samples were analyzed on a NanoLC-ESI-MS/MS system (Q-Exactive Plus Quadrupole Orbitrap) (Thermo Fisher Scientific) coupled to a Agilent 1100 HPLC system (Agilent), a service provided by Prottech, Phoenixville, PA. In detail, DQβ0603:DQα0103 antigens obtained from $2 \times 10^9$ cells in 10% acetic acid and 0.1% TFA solution were treated at 90 °C for 40 min, cooled down to room temp before freezing down at −80 °C for 20 min. Thawed sample were spin at 8000 rpm 20 min and filtered with 2 micrometer (µm) glass filter to remove protein aggregates. Samples were loaded on an 1 ml C18 packed columns (Pierce spin column, Thermo Fisher Scientific, cat # 89870), washed with 20 ml 0.1% TFA and HLA restricted peptides were eluted with 2 ml 50% ACN in 0.1% TFA.

NanoLC-ESI-MS/MS analysis of peptide samples were carried out by a high pressure liquid chromatography (HPLC) system (Agilent) coupled with a 75 µm inner diameter 10 cm in length reverse phase C18 column (beads from Thermo Fisher Scientific, 3 µm particle size, 300 Å pore size). Samples (3 µl) were loaded at flow rate of 800 nanoliter per minute. After washing the C18 column for 20 min with buffer A (97.5%

water, 2% acetonitrile, 0.5% formic acid), the DQβ0603:DQα0103-associated peptides were separated over an increasing gradient of buffer B (9.5% water, 90% acetonitrile, 0.5% formic acid) in 1 hour.

Peptides eluted from HPLC C18 column were directly ionized by ion trap mass spectrometer (QE, Thermo Fisher Scientific) with an ionization voltage at 2.4kv at 120 °C. Data were collected in data-dependent mode with normalized collision energy (NCE) set at 27%. One full scan with 2 microscan with a mass range of 350 amu to 2000 amu was acquired, followed by one MS/MS scan of the most intense ion with a full mass range and two microscans. The dynamic exclusion feature was set to be 30 seconds.

The mass spectrometric data was used to search against the most recent non-redundant protein database (NR database) from NCBI with ProtTech's ProtMody software suite by parameters of the data acquisition: instrument: LTQ; software: Xcalibur 2.0; centroid mode; minimum MS signal for Precursor-ion: 5×104 counts. For the processing parameters settings, ProtMody 2.0 Software was used with the settings of: signal-to-noise: >=5; de-isotoped: yes; filtered: yes; modifications: Cys 0; database: protein database from UniProt Database.

### Peptide loading protocol

HLA Peptides were synthesized by ThermoFisher Scientific (Waltham, MA). Biotinylated HLA peptides (0.5 mM) were incubated with approximately 7,500 beads bearing DQβ0601:DQα0103 protein purified from the BLS transfectant in 50 μl of 150 mM citrate phosphate buffer pH7.2 for 72 hours at 37 °C. The reaction mixture was then centrifuged at 14,000 x g to pellet the beads, after removing the supernatant, beads were washed twice with 300 μl LABScreen wash buffer and resuspended in 25 μl phosphate buffered saline solution (Irvine Scientific, Santa Ana, CA) with 2% BSA (Sigma Aldrich, Burlington, MA). Serum reactivity was then measured according to LABScreen SAB assay protocol.

### Statistics

Statistical significance was determined using one-way analysis of variance (ANOVA) followed by Bonferroni post hoc test or using Student's t test in Microsoft® Excel® for Microsoft 365 MSO (Version 2311 Build 16.0.17029.20028) as indicated in figure legends. The statistical analysis results are provided in Source data file.

### Reporting summary

Further information on research design is available in the Nature Portfolio Reporting Summary linked to this article.

## Data availability

The source data and the statistical analysis results are provided with this article in Source data file. Other source data that are not in the Source data file can be obtained from the corresponding author (J.-H.L.) by reasonable request. The peptidomics dataset "MHC peptidome analysis of HLA DQβ0603:DQα0103 proteins has been deposited in PRIDE under accession code PXD043999 (http://www.ebi.ac.uk/pride/archive/projects/PXD043999). Source data are provided with this paper.

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

## Acknowledgements
This work was supported by research funds provided by the Paul I. Terasaki Research Fund and the Transplant Diagnostics Business of Thermo Fisher Scientific. We thank Dr. Mark Terasaki for his constructive discussion.

## Author contributions
J.-H.L., N.R.S. and T.N. designed the study, collected the data, analyzed the data and interpreted the data. R.A.B., C.M., J.-L.T., M.D. and J.N. screened, analyzed and identified study samples. M.-B.M. evaluated and interpreted the clinical data. J.-H.L., R.A.B., J.N., J.-L.T., M.D. and M.L.-C. drafted the article and revised the article. R.A.B., C.M., M.L.-C. and P.W.N analyzed the data, interpreted the data and revised the article.

## Competing interests
T.N., N.R.S., M.L.-C., J.-L.T., M.D., J.N. and M.-B.M. have no conflicts of interest to disclose. J.-H.L. is a consultant for the Transplant Diagnostics Business of Thermo Fisher Scientific. P.W.N. is a consultant for CSL Behring. C.M. is on the Advisory Board of Luminex Corporation. R.A.B. is on the Advisory Board of Luminex Corporation.
