## [Peer Review File · Nature Communications]

REVIEWER COMMENTS

Reviewer #1 (expert in HLA-specific antibodies and transplantation):

This is a well written paper with implications for how HLA immunization is interpreted in the transplant setting, and most importantly, how the virtual crossmatch is applied. It has been known for some time that cytotoxic T cell specificity can be peptide dependent but as you claim this is the first clear demonstration of peptide dependent antibody activity.

I feel that this MS has the potential to change thinking on the subject of antibody reactivity, if it turns out with further investigation to be a common feature. As it stands it would appear that HLA class I of the recipient has to be taken into account in the presence of the DQ dimer studied. In fact, other non- HLA polymorphic genes may have to be considered if this phenomenon is widespread.

I have two questions for the authors.

1. Barbara Nepom showed years ago that DQ dimers can exist in the trans as well as in a cis configuration. While this could add a further level of complexity will the authors take this into consideration when looking at other DQ specificities?

2. Could this observed phenomenon also apply to antibodies directed at DRB1/3/4 and DPB1/DPA1?

Reviewer #2 (expert in HLA immunogenetics and structure):

Review manuscript NCOMMS-23-31943-T

The manuscript titled: HLA Class I Peptide Polymorphisms Contribute to the Class II DQ β 0603:DQ α 0103 Antibody Specificity by R. Shih et al., is an interesting study addressing an important point related to the interpretation of SAB data generated using patients' sera and DQ-carrying beads. The basic finding, as presented by the authors, is that this is the first report implying that peptides, bound to HLA Class II proteins, can be part of the antibody binding epitope.

Broad comments: Even though the presented data convincingly make the point of the influence of specific peptides to the reactivity of serum anti-DQ antibodies with the DQ-peptide complex, the concept itself, regarding the influence of peptide binding to HLA molecules, whether class I or class II, as they react with Abs (monoclonal or serum Abs) or even T-cell receptors (TCR), is not entirely new. The demonstration of the specific DQ with the specific peptides is new, but the concept is not. Below, please find a number of references making similar claims since early 1990s:

Catipovic B. et al. Major Histocompatibility Complex Conformational Epitopes Are Peptide Specific J. Exp. Med., 1992, 176: 1611-1618.

Bluestone J. et al. Peptide-induced Conformational Changes in Class I Heavy Chains Alter Major Histocompatibility Complex Recognition. J. Exp. Med., 1992, 176: 1757-1761

Catipovic B. et al. Analysis of the Structure of Empty and Peptide-loaded Major Histocompatibility Complex Molecules at the Cell Surface J. Exp. Med., 1994, 180: 1753-1761

Mulder A. et al. Impact of Peptides on the Recognition of HLA Class I Molecules by Human HLA Antibodies. The Journal of Immunology, 2005, 175: 5950-5957.

All the above examples, admittedly, involve class I molecules, but the concept is the same.

Furthermore, the following reference is about the reactivity of DQ molecules as they interact with monoclonal Abs:

Monos D. et al. L cells expressing DQ molecules of the DR3 and DR4 haplotypes: reactivity with mAbs. Immunogenetics, 1995, 42: 172-180.

There is hypothesized that the reactivity patterns can be explained by the influence of different peptides bound to DQ molecules.

Interestingly the following reference is a detailed crystallographic study involving the class I molecule B8 as it interacts with different peptides and the HLA-peptide complexes interacting with a single TCR. Obviously, a TCR can be like the variable site of an Ab and as such the conclusions of the study can be relevant to the DQ-peptide-Ab trimolecular complex.

Reid S. et al. Antagonist HIV-1 Gag Peptides Induce Structural Changes in HLA B8. J. Exp. Med., 1996, 184: 2279-2286.

The major contribution of the above paper is that because of the detailed crystallographic analysis the differences in reactivities are attributed to at least three different ways that the bound peptides influence the reactivity with the TCR. The statement made by the authors "peptides, bound to HLA Class II proteins, can be part of the antibody binding epitope" is not necessarily accurate. The peptide can influence the reactivity of the HLA-peptide complex with an Ab or TCR in more than one way. The peptides do not have to be part of the Ab binding epitope to influence reactivity with the Ab. I would strongly urge the authors to review the above papers and account for their content, as they set the frame for their study.

While the SAB assay is of undisputable significance and impact for the transplantation field, there are several aspects of this assay that remain unclear, occasionally challenging lab interpretations for patient care. Therefore, this work is very important to the field and all the comments below aim at improving presentation and interpretation.

Regarding the presentation of the data, the text is sometimes confusing and difficult to follow. It would help if the authors included a sentence or two as they describe a new line of experiments, explaining the rationale and the intended purpose so the reader makes the proper connections. The list of experiments without a comprehensive underlying connecting rationale makes it difficult to follow.

In "Methods" it would be meaningful to list the cell line that was used to generate the DQ β 0603:DQ α 0103 bead that was used to identify the 21 sera. It may be somehow relevant in interpreting some of the subsequent data. Were all sera identified of comparable reactivity with this single bead?

Assessing reactivity patterns appears to be from a single replicate of each experiment. I am wondering of its reproducibility. Were the experiments done more than once? Please comment. I would consider adequate to repeat the SAB assays not necessarily the making of the SABs. Line 148: "Serum S6 has weaker reactivity to the DQ β 0603:DQ α 0103 protein derived from 9060 and 9065 than the other 3 proteins derived from 9058, 9062 and 9105". From Figure 1a, it looks like there is maybe a difference of 2000 MFI b, between the reactivity of 9058 and 9060. Is this really a significant difference when you are already above an MFI of 10000? I would agree with the assertion that there is a significant difference in S1 with cell lines 9058 and 9062 compared to the other three, but most of the differences in S2 - S6 could be due to natural variation in the system. This is difficult to tease out without having replicates of the experiment to account for the natural variation. Also, all sera had a lower reactivity to the peptides from 9065 than the peptides from all other cell lines, so I'm not sure that this is any substantial observation with the S6 sera. With regards to the peptides derived from 9065, I would rethink how there may be a difference in the preparation of the peptides from this cell line that is causing the "weaker reactivity".

Technical issues with the transfectants expressing DQ β 0603:DQ α 0103. Do we have any controls demonstrating that these beads express comparable levels of DQ β 0603:DQ α 0103? Irrespectively of the reactivity of these beads with the sera, the total amount of DQ molecules needs to be the same in all beads. The authors have shown that there is a positive control for all DQs, the DQ6PC in Fig 1a or FM5148 used for purifying the DQs from the transfectants. Why not the same here? Did the authors considered the testing of the stable transfectants of DQ β 0603:DQ α 0103 with the same 6 sera directly, instead of going through SAB? Was there an issue with level of expression of the DQ β 0603:DQ α 0103 on the different transfectants? I am familiar with the challenges of generating stable DQ transfectants. Manipulating the DQs adds another level of complexity. I am asking the question, I do not suggest the experiment.

Line 179: It is stated that "there was yet another mechanism responsible for the DQ β 0603:DQ α 0103 antibody reactivity" The authors never come back to elaborate on this comment and assess what would be the impact on other results if this was the case. May be a comment in the "Discussion"?

Table 1: Make it more informative by including information of expressing class I or class II. The info is in Fig 2 but there is confusion with the designation of the star (*). Is one star no expression of class II only? Also, the listing of the HLA class I alleles in Table 1 creates the impression of expression as well. Clarify and distinguish the HLA genotyping from its expression of class I. Difficult to follow, as the info is distributed in Methods, Fig 2 and Table 1. Also, why the C*01:02 is

included in the Table 1 for LCLKO, if this cell line is derived from LCL721.221 that lacks expression of all class I? The experiments involving co-transfection of LCLKO cells with the DQ β 0603:DQ α 0103 and class I A2 and C16 suggests that this is most likely an oversight. Unclear as to whether the LCLKO has been co-transfected with both A2 and C16 or separately. The text states: HLA A*02:01 and C*16:01 alleles from 9058 cell line (Table 1), were introduced into the LCLKO transfectant expressing DQ β 0603:DQ α 0103 antigen. Is it "and" or it is "or"? The logic of the experiment suggests that is "or" not "and". The legend of Fig 3 states "or".

While this work is interesting and relevant to the operations of clinical laboratories, how do the authors realistically see investigating these odd reactivity patterns that may be due to the peptide presented as part of the MHC Class II specificities on the single antigen beads during for routine clinical work? The work presented here does not lend itself to day to day operations in a clinical lab and routine patient evaluation.

Related questions: What proportion of DQ β 0603:DQ α 0103 sera show peptide dependent reactivity? Do we expect that all sera with this specificity will have this peptide dependency? How prevalent is this in other class IIs? What about other DQs such as DQA1*01:02:DQB1*06:02 which is very similar in sequence?

What are the backgrounds of the 21 patients/sera donors? Do they have documented exposure to the DQ in question?

Dimitri Monos

Reviewer #3 (expert in immunogenetics and transplantation):

The manuscript submitted by Shih et al. aimed to demonstrate how the peptides of polymorphic HLA class I heavy chain loaded in the DQ β 0603-DQ α 0103 molecule contribute to DQ β 0603-DQ α 0103 antibody specificity. This is an interesting effort, but the findings were based solely on solid-phase HLA antibody assay, which is often unverified by cell-based crossmatch assay. I have the following critiques.

Major concerns:

1. The investigators created a panel of Luminex beads coated with DQ β 0603:DQ α 0103 proteins purified from five homozygous EBV-transformed B cell lines with different HLA class I genotypes. They tested DQ β 0603:DQ α 0103 specific sera isolated from six patients with this bead panel. Among the three sera (Fig.1a: S2, S3, S4), there were similar reactivity patterns across all 6 beads. However, the reactivity of the other 3 sera varied among the beads, and this variation was correlated with the HLA class I type of the cell line from which the DQ β 0603:DQ α 0103 was isolated. For Serun#1, it only bound to DQ β 0603:DQ α 0103 which was isolated from HLA-A2 positive B cell lines. The other 2 sera (S5, S6) showed subtle differences in MFI among the five beads, which could be due to the well-known Luminex inter-assay MFI variation (Reed et al. Am J Transplant. 2013, 13:1859-70). Therefore, please provide the mean and SD of multiple experiments to differentiate between the real reactivity and intra-/inter-assay MFI variations.
2. The reactivities of 4 DQ α /DQ β monoclonal antibodies (mAb) to the DQ β 0603:DQ α 0103 bead panel (Fig.1b) were at the higher end, ranging from 332,000 MFI to 45,000 MFI. However, the reactivities of the 6 patient sera and DQ6-PC range only from 5,000 to 25,000 MFI (Fig.1a). It is expected that DQ6-PC should react at similar MFI levels compared to mAbs. Can you please explain the reason for such a big difference between mAb and patient serum reactivities?
3. The data presented here indicate that DQ β 0603:DQ α 0103 molecules present peptides of the cognate HLA-A2 heavy-chain, expressed on the same EBV-transformed B cell lines. Presenting self-HLA class I heavy-chain-derived peptides by DQ molecules challenges the current dogma. Does this mean that healthy individuals having DQ β 0603:DQ α 0103 and HLA-A2 should be generating antibodies against DQ β 0603:DQ α 0103? Do you have data supporting this? Moreover, how does this self-class I peptide presentation by DQ relate to the allogeneic transplant setting? Do you have transplant recipient data showing that donor HLA-A2 (but not A1/A3) heavy-chain peptides are presented by recipient DQ β 0603:DQ α 0103, thereby triggering DQ β 0603:DQ α 0103 antibody production?

4. Table 2 lists HLA class I peptides identified from the DQ β 0603:DQ α 0103 protein peptidome analysis. The HLA-A*02:01 heavy-chain peptide carrying the epitope (aa 142TTKH145) is identified as the determinant of the DQ β 0603:DQ α 0103 antibody specificity in Serum S1. This epitope aa-142TTKH145 is also shared by A68 and A69. Do you have data from cell lines expressing DQ β 0603:DQ α 0103 and HLA-A68/A69?

5. The cell surface expression of HLA class I proteins is down-regulated in EBV-transformed B cell lines compared to normal cells (Dutta et al. Cancer 2006, 106:1685-93). Defects in the class I-associated antigen processing and presentation pathway, particularly the downregulation of the transporter associated with antigen processing (TAP), cause reduced HLA class I on the cell surface (Khanna et al., Int Immunol. 1994, 6:639-45). A failed TAP system likely accumulates class I proteins within the cytoplasm, and such an aberrant accumulation of abundant class I chains may lead to unusual DQ-presentation of self-HLA class I proteins. Therefore, the finding of this study is more likely a phenomenon specific to EBV-transformed B cell lines. A new experimental design is necessary to assess the allogeneic class I peptide presentation by DQ molecules to suggest the relevance of this research in transplantation.

6. In the introduction, the authors clearly outlined the gap in the field - the inconsistency between solid-phase assay antibody results and cell-based cross-matches, causing inefficiency in organ allocation and ABMR diagnosis/treatment. However, the present study completely ignored such fundamental experiments of cell-binding assays. Have you performed a cell-based crossmatch of all 6 sera with DQ β 0603:DQ α 0103 antibody? In particular, to confirm serum#1 reactivity in Fig.1a, it will be interesting to see a positive crossmatch with a target carrying HLA-A2+DQ β 0603:DQ α 0103 but negative with the targets carrying HLA-A1/A3+DQ β 0603:DQ α 0103. In our experience, DQ β 0603:DQ α 0103 antibodies do not bind to cells despite MFI. If they do not bind to cells, the downstream function and clinical utility of DQ β 0603:DQ α 0103 are questionable. Therefore, please provide cell-binding data and possibly transplants showing relevance in ABMR.

7. Fig.2: Please provide the control reactivities (i.e., DQ6-PC and DQ α /DQ β mAbs) of recombinant DQ β 0603:DQ α 0103 proteins extracted from six stable transfectants of CD74 and DM positive host cell lines. What is the impact of HLA class I genotypes on these reactivities? Why is S6 completely negative with all 6 beads?

Minor Comments:

1. What was the purity and yield of peptides? The HPLC report is missing in the manuscript. Electrospray Ionization (ESI) histograms that generally show the presence of peptides detected by mass spectrometry are also missing.

2. It is confusing to see the amino acid positions noted using "codon," which is meant for the nucleotide sequence.

3. Page 15, line no. 264: replace "demostrated" with "demonstrated."

4. The references are not in one format. For example, some references have a full stop at the end but others do not.

RESPONSE TO REVIEWERS' COMMENTS

We thank the three reviewers for their valuable comments.

We have tried to address most of, if not all, the comments the reviewers brought up. We believe by addressing those comments we have improved the quality of this manuscript significantly.

Below are our responses to each comment in green.

The line numbers used in the response refer to the main text file without track changes.

Reviewer #1 (expert in HLA-specific antibodies and transplantation):

This is a well written paper with implications for how HLA immunization is interpreted in the transplant setting, and most importantly, how the virtual crossmatch is applied. It has been known for some time that cytotoxic T cell specificity can be peptide dependent but as you claim this is the first clear demonstration of peptide dependent antibody activity.

I feel that this MS has the potential to change thinking on the subject of antibody reactivity, if it turns out with further investigation to be a common feature. As it stands it would appear that HLA class I of the recipient has to be taken into account in the presence of the DQ dimer studied. In fact, other non-HLA polymorphic genes may have to be considered if this phenomenon is widespread.

Thank you for the insight. We have added a comment to the discussion (Line 328).

I have two questions for the authors.

1. Barbara Nepom showed years ago that DQ dimers can exist in the trans as well as in a cis configuration. While this could add a further level of complexity will the authors take this into consideration when looking at other DQ specificities?

This is an important point and in future experiments, we will try to create heterodimers with uncommon DQA1 and DQB1 combinations to check the serum reactivity if those combinations can produce stable transfectants.

2. Could this observed phenomenon also apply to antibodies directed at DRB1/3/4 and DPB1/DPA1?

Interesting point. DR and DP have similar binding structure as DQ, so, in theory, a similar phenomenon could exist. Clearly, this concept needs further investigation.

Reviewer #2 (expert in HLA immunogenetics and structure):

Review manuscript NCOMMS-23-31943-T

The manuscript titled: HLA Class I Peptide Polymorphisms Contribute to the Class II DQ β 0603:DQ α 0103 Antibody Specificity by R. Shih et al., is an interesting study addressing an important point related to the interpretation of SAB data generated using patients' sera and DQ-carrying beads. The basic finding, as presented by the authors, is that this is the first report implying that peptides, bound to HLA Class II proteins, can be part of the antibody binding epitope.

Broad comments: Even though the presented data convincingly make the point of the influence of specific peptides to the reactivity of serum anti-DQ antibodies with the DQ-peptide complex, the concept itself, regarding the influence of peptide binding to HLA molecules, whether class I or class II, as they react with Abs (monoclonal or serum Abs) or even T-cell receptors (TCR), is not entirely new. The demonstration of the specific DQ with the specific peptides is new, but the concept is not. Below, please find a number of references making similar claims since early 1990s:

Catipovic B. et al. Major Histocompatibility Complex Conformational Epitopes Are Peptide Specific J. Exp. Med., 1992, 176: 1611-1618.

Bluestone J. et al. Peptide-induced Conformational Changes in Class I Heavy Chains Alter Major Histocompatibility Complex Recognition. J. Exp. Med., 1992, 176: 1757-1761

Catipovic B. et al. Analysis of the Structure of Empty and Peptide-loaded Major Histocompatibility Complex Molecules at the Cell Surface J. Exp. Med., 1994, 180: 1753-1761

Mulder A. et al. Impact of Peptides on the Recognition of HLA Class I Molecules by Human HLA Antibodies. The Journal of Immunology, 2005, 175: 5950–5957.

All the above examples, admittedly, involve class I molecules, but the concept is the same. Furthermore, the following reference is about the reactivity of DQ molecules as they interact with monoclonal Abs:

Monos D. et al. L cells expressing DQ molecules of the DR3 and DR4 haplotypes: reactivity with mAbs. Immunogenetics, 1995, 42: 172-180.

There is hypothesized that the reactivity patterns can be explained by the influence of different peptides bound to DQ molecules.

Interestingly the following reference is a detailed crystallographic study involving the class I molecule B8 as it interacts with different peptides and the HLA-peptide complexes interacting with a single TCR. Obviously, a TCR can be like the variable site of an Ab and as such the conclusions of the study can be relevant to the DQ-peptide-Ab trimolecular complex.

Reid S. et al. Antagonist HIV-1 Gag Peptides Induce Structural Changes in HLA B8. J. Exp. Med., 1996, 184: 2279-2286.

The major contribution of the above paper is that because of the detailed crystallographic analysis the differences in reactivities are attributed to at least three different ways that the bound peptides influence the reactivity with the TCR. The statement made by the authors “peptides, bound to HLA Class II proteins, can be part of the antibody binding epitope” is not necessarily accurate. The peptide can influence the reactivity of the HLA-peptide complex with an Ab or TCR in more than one way. The peptides do not have to be part of the Ab binding epitope to influence reactivity with the Ab. I would strongly urge the authors to review the above papers and account for their content, as they set the frame for their study.

We have amended the wording to clarify the difference between our findings and previous literatures, see Line 30 and 326.

While the SAB assay is of undisputable significance and impact for the transplantation field, there are several aspects of this assay that remain unclear, occasionally challenging lab interpretations for patient care. Therefore, this work is very important to the field and all the comments below aim at improving presentation and interpretation.

Regarding the presentation of the data, the text is sometimes confusing and difficult to follow. It would help if the authors included a sentence or two as they describe a new line of experiments, explaining the rationale and the intended purpose so the reader makes the proper connections. The list of experiments without a comprehensive underlying connecting rationale makes it difficult to follow.

We have added the new statements to help clarify the transition between sections.

In “Methods” it would be meaningful to list the cell line that was used to generate the DQβ0603:DQα0103 bead that was used to identify the 21 sera. It may be somehow relevant in interpreting some of the subsequent data. Were all sera identified of comparable reactivity with this single bead?

The DQβ0603:DQα0103 positive sera were first identified via routine screening using the commercial HLA Class II Single Antigen kit at each transplant center participating in this study. The MFI ranges of these 21 sera were between 3,000-50,000. The assay was done at different times, using different lots of beads, different ancillary reagents and different Luminex instruments, either LABScan100 or LABScan3D. A minimum cut-off of 3000 MFI was added to the methods to clarify the selection criteria, see Line 374.

Assessing reactivity patterns appears to be from a single replicate of each experiment. I am wondering of its reproducibility. Were the experiments done more than once? Please comment. I would consider adequate to repeat the SAB assays not necessarily the making of the SABs.

The SAB assays in Figure 1-4 were repeated at least two times using different lots of secondary antibody and different Luminex FM3D instruments. For the revised figures, we have repeated the experiments in triplicate using the same secondary antibody and same instrument for better consistency.

Line 148: “Serum S6 has weaker reactivity to the DQβ0603:DQα0103 protein derived from 9060 and 9065 than the other 3 proteins derived from 9058, 9062 and 9105”. From Figure 1a, it looks like there is maybe a difference of 2000 MFI, between the reactivity of 9058 and 9060. Is this really a significant difference when you are already above an MFI of 10000? I would agree with the assertion that there is a

significant difference in S1 with cell lines 9058 and 9062 compared to the other three, but most of the differences in S2 – S6 could be due to natural variation in the system. This is difficult to tease out without having replicates of the experiment to account for the natural variation. Also, all sera had a lower reactivity to the peptides from 9065 than the peptides from all other cell lines, so I'm not sure that this is any substantial observation with the S6 sera. With regards to the peptides derived from 9065, I would rethink how there may be a difference in the preparation of the peptides from this cell line that is causing the "weaker reactivity".

In our experience, and as observed with the small variation in the new repeated measurements, many of the smaller differences are likely to be real differences and not related to normal variability of the assay. We agree however that this is a valid point and we have modified the description for sera S2-S6 without emphasizing those small MFI variations, see Line112.

Technical issues with the transfectants expressing DQ β 0603:DQ α 0103. Do we have any controls demonstrating that these beads express comparable levels of DQ β 0603:DQ α 0103? Irrespectively of the reactivity of these beads with the sera, the total amount of DQ molecules needs to be the same in all beads. The authors have shown that there is a positive control for all DQs, the DQ6PC in Fig 1a or FM5148 used for purifying the DQs from the transfectants. Why not the same here?

Thanks for this important point, we have added the reactivity of Control monoclonal antibodies to Figure 2 and 3.

Did the authors considered the testing of the stable transfectants of DQ β 0603:DQ α 0103 with the same 6 sera directly, instead of going through SAB? Was there an issue with level of expression of the DQ \square 0603:DQ \square 0103 on the different transfectants? I am familiar with the challenges of generating stable DQ transfectants. Manipulating the DQs adds another level of complexity. I am asking the question, I do not suggest the experiment.

The expression level of DQ β 0603:DQ α 0103 protein on the transfectants varies. LCL3023, T2DM and LCLKO which are expressing CD74 and DM had higher surface expression than BLS, K562 or T2 which do not. With the purified antigens, we can adjust the amount of antigen on the beads to a similar level for better comparison. The Flow results of S1, the strongest serum in Figure 2a, have been added in Figure 2c and 2d.

Line 179: It is stated that "there was yet another mechanism responsible for the DQ β 0603:DQ α 0103 antibody reactivity" The authors never come back to elaborate on this comment and assess what would be the impact on other results if this was the case. May be a comment in the "Discussion"?

We have clarified the language, see Line 168-171.

Table 1: Make it more informative by including information of expressing class I or class II. The info is in Fig 2 but there is confusion with the designation of the star (*). Is one star no expression of class II only? Also, the listing of the HLA class I alleles in Table 1 creates the impression of expression as well. Clarify and distinguish the HLA genotyping from its expression of class I. Difficult to follow, as the info is distributed in Methods, Fig 2 and Table 1. Also, why the C*01:02 is included in the Table 1 for LCLKO, if this cell line is derived from LCL721.221 that lacks expression of all class I? The experiments involving co-transfection of LCLKO cells with the DQ β 0603:DQ α 0103 and class I A2 and C16 suggests that this is most likely an oversight.

The parent of LCLKO, LCL721.221, has a very low level expression of C*01:02 (Partridge T, Nicastri A, Kliszczyk AE, Yindom LM, Kessler BM, Ternette N, Borrow P. Discrimination between human leukocyte antigen Class I-Bound and Co-Purified HIV-Derived peptides in immunopeptidomics workflows. *Front Immunol.* 2018;9:912. doi: 10.3389/fimmu.2018.00912.). However, the DQB0603:DQα0103 protein purified from LCLKO was not reactive to the six sera in this study. In order to clarify this, columns for HLA Class I and II expression of the cell lines were added to Table 1.

Unclear as to whether the LCLKO has been co-transfected with both A2 and C16 or separately. The text states: HLA A*02:01 and C*16:01 alleles from 9058 cell line (Table 1), were introduced into the LCLKO transfectant expressing DQB0603:DQα0103 antigen. Is it “and” or it is “or”? The logic of the experiment suggests that is “or” not “and”. The legend of Fig 3 states “or”.

Thank you for pointing this out we have changed the word “and” to “or” on Line 176.

While this work is interesting and relevant to the operations of clinical laboratories, how do the authors realistically see investigating these odd reactivity patterns that may be due to the peptide presented as part of the MHC Class II specificities on the single antigen beads during for routine clinical work? The work presented here does not lend itself to day to day operations in a clinical lab and routine patient evaluation.

In contrast, we think our work may be relevant to day-to-day practice. The generalizability of our findings of course needs to be investigated with regards to other HLA-antigens and serum reactivity based on different Class I background of the recombinant antigens. We do not know how common this phenomenon is mainly because this is as of yet not a well-known or published observation. Hypothetically, a portion of ABMR positive, DSA Negative cases may have been due to such peptide dependent antibodies that went unappreciated. Informing the community of such possibilities is an important first step, which then has to be followed by more research. With the addition of case S22, which demonstrated de novo DQB0603:DQα0103 DSA after transplantation with clinical AMR we also show the possible clinical relevance of the detected DQB0603:DQα0103 targeting antibodies described in this manuscript.

Related questions: What proportion of DQB0603:DQα0103 sera show peptide dependent reactivity? Do we expect that all sera with this specificity will have this peptide dependency?

We only selected the sera that reacted to this bead and not other DQB1*06 or DQB1*05 beads for this study, all 22 sera that were selected with these criteria have this peptide dependency. A paragraph was added to the discussion to clarify this. While we didn't select sera that had more than one DQ1 beads positive, it would be very interesting to look at the peptide dependency in sera with broader reactivity.

How prevalent is this in other class IIs? What about other DQs such as DQA1*01:02:DQB1*06:02 which is very similar in sequence?

This is a very interesting question and we will expand our future planned studies to include other DQ specificities.

What are the backgrounds of the 21 patients/sera donors? Do they have documented exposure to the DQ in question?

Among the 21 patients, 12 had been transplanted with preformed DQ β 0603:DQ α 0103 non-DSA antibody, the other 9 are still awaiting transplant. Most of the 21 patients had no documented sensitization event when the antibody was first detected. We have included an additional patient in our revised submission that received a DQB1*06:03, DQA1*01:03 mismatched kidney and developed DSA to this antigen to highlight that this could indeed be clinically relevant antibodies.

Dimitri Monos

Reviewer #3 (expert in immunogenetics and transplantation):

The manuscript submitted by Shih et al. aimed to demonstrate how the peptides of polymorphic HLA class I heavy chain loaded in the DQ β 0603-DQ α 0103 molecule contribute to DQ β 0603-DQ α 0103 antibody specificity. This is an interesting effort, but the findings were based solely on solid-phase HLA antibody assay, which is often unverified by cell-based crossmatch assay. I have the following critiques.

Major concerns:

1. The investigators created a panel of Luminex beads coated with DQ β 0603:DQ α 0103 proteins purified from five homozygous EBV-transformed B cell lines with different HLA class I genotypes. They tested DQ β 0603:DQ α 0103 specific sera isolated from six patients with this bead panel. Among the three sera (Fig.1a: S2, S3, S4), there were similar reactivity patterns across all 6 beads. However, the reactivity of the other 3 sera varied among the beads, and this variation was correlated with the HLA class I type of the cell line from which the DQ β 0603:DQ α 0103 was isolated. For Serun#1, it only bound to DQ β 0603:DQ α 0103 which was isolated from HLA-A2 positive B cell lines. The other 2 sera (S5, S6) showed subtle differences in MFI among the five beads, which could be due to the well-known Luminex inter-assay MFI variation (Reed et al. Am J Transplant. 2013, 13:1859-70). Therefore, please provide the mean and SD of multiple experiments to differentiate between the real reactivity and intra-/inter-assay MFI variations.

The SAB experiments in Figure 1-4 were repeated at least two times using different lots of secondary antibody and different Luminex FM3D instruments. For the revised figures, we have repeated the experiments in triplicate using the same secondary antibody and same instrument for better consistency. This is visible in the new Figures included in the revised submission.

2. The reactivities of 4 DQ α /DQ β monoclonal antibodies (mAb) to the DQ β 0603:DQ α 0103 bead panel (Fig.1b) were at the higher end, ranging from 332,000 MFI to 45,000 MFI. However, the reactivities of the 6 patient sera and DQ6-PC range only from 5,000 to 25,000 MFI (Fig.1a). It is

expected that DQ6-PC should react at similar MFI levels compared to mAbs. Can you please explain the reason for such a big difference between mAb and patient serum reactivities?

We agree that the range of the monoclonal antibodies is high (30 000 – 45 000 MFI) one reason is that mAbs usually are selected for high affinity binding. So compared to patient sera, it would be expected that they may have higher MFIs. The second reason is that the mAbs used were developed to recognize a sequence-specific structure of either the DQA1 or DQB1 proteins, so all DQ molecules on the beads are capable of binding to the mAbs. The sera we tested recognize an epitope that involve a specific peptide which representing only a fraction of the DQ molecules bound to the beads. We have added an additional positive control serum with higher reactivity that has similar MFI as the mAbs which is shown in the revised Figure 1b.

3. The data presented here indicate that DQ β 0603:DQ α 0103 molecules present peptides of the cognate HLA-A2 heavy-chain, expressed on the same EBV-transformed B cell lines. Presenting self-HLA class I heavy-chain-derived peptides by DQ molecules challenges the current dogma. Does this mean that healthy individuals having DQ β 0603:DQ α 0103 and HLA-A2 should be generating antibodies against DQ β 0603:DQ α 0103? Do you have data supporting this? Moreover, how does this self-class I peptide presentation by DQ relate to the allogeneic transplant setting? Do you have transplant recipient data showing that donor HLA-A2 (but not A1/A3) heavy-chain peptides are presented by recipient DQ β 0603:DQ α 0103, thereby triggering DQ β 0603:DQ α 0103 antibody production?

All 21 patients were from patients either have received or awaiting kidney transplant. We don't know whether healthy individuals will produce this type of antibody or not since we have not investigated this. We have added data from a patient producing DQ β 0603:DQ α 0103 DSA 6 years post kidney transplant in the Results section and discussed the potential immunogenicity in the Discussion section. We didn't have data to suggest the scenario Reviewer #3 mentioned that "donor HLA-A2 (but not A1/A3) heavy-chain peptides are presented by recipient DQ β 0603:DQ α 0103, thereby triggering DQ β 0603:DQ α 0103 antibody production". These questions need to be investigated in the future with additional experiments and screening of larger cohorts.

4. Table 2 lists HLA class I peptides identified from the DQ β 0603:DQ α 0103 protein peptidome analysis. The HLA-A*02:01 heavy-chain peptide carrying the epitope (aa 142TTKH145) is identified as the determinant of the DQ β 0603:DQ α 0103 antibody specificity in Serum S1. This epitope aa-142TTKH145 is also shared by A68 and A69. Do you have data from cell lines expressing DQ β 0603:DQ α 0103 and HLA-A68/A69?

Unfortunately, we don't have DQ β 0603:DQ α 0103 and A*68/*69 co-expressing cell line in our collection. However, from the peptide loading experiment (Fig 5a), we would expect any HLA allele that carries the 142TTKH145 motif would be positive with Serum S1.

5. The cell surface expression of HLA class I proteins is down-regulated in EBV-transformed B cell lines compared to normal cells (Dutta et al. Cancer 2006, 106:1685-93). Defects in the class I-associated antigen processing and presentation pathway, particularly the downregulation of the transporter associated with antigen processing (TAP), cause reduced HLA class I on the cell surface (Khanna et al., Int Immunol. 1994, 6:639-45). A failed TAP system likely accumulates class I proteins within the cytoplasm, and such an aberrant accumulation of abundant class I chains may lead to unusual DQ-presentation of self-HLA class I proteins. Therefore, the finding of this study is more likely a phenomenon specific to EBV-transformed B cell lines. A new experimental design is necessary to assess the allogeneic class I peptide presentation by DQ molecules to suggest the relevance of this research in transplantation.

The B cell lines used in our study were extensively characterized in the 10th International Histocompatibility Workshop using complement-dependent cytotoxicity assay with a large panel of human sera (Serologic Characterization of the Reference Panel of B-Lymphoblastoid Cell Lines for Factors of the HLA System. E.L. Milford, LJ Kennedy, SY Yang, B Dupont, J-M Lalouel and EJ Yunis. Immunology of HLA 1987:19-38. The surface expression of HLA Class I proteins on those cell lines were all confirmed, so it's unlikely that the B cell lines we used have over accumulated Class I proteins inside the cells due to the deficiency in TAP system as the reported EBV infected gastric cancer cells (Dutta et al. Cancer 2006, 106:1685-93).

Beside the observation of EBV transformed B-cell lines containing the HLA Class I peptide (Figure 1a), we also have in-vitro peptide loading experiments independently showing by loading the peptides with the same sequences as we identified from the B cell lines had the same effect (Figure 4), so it is unlikely to be a phenomenon specific to EBV-transformed B cell lines. Regardless of whether B cell line accumulate HLA Class I peptides or not, we demonstrate kidney transplant patients produced such antibodies. We have added a section investigating an allogenic DQβ0603:DQα0103 specific DSA antibody produced in a patient that had received a DQβ0603:DQα0103 graft.

5. In the introduction, the authors clearly outlined the gap in the field - the inconsistency between solid-phase assay antibody results and cell-based cross-matches, causing inefficiency in organ allocation and ABMR diagnosis/treatment. However, the present study completely ignored such fundamental experiments of cell-binding assays. Have you performed a cell-based crossmatch of all 6 sera with DQβ0603:DQα0103 antibody? In particular, to confirm serum#1 reactivity in Fig.1a, it will be interesting to see a positive crossmatch with a target carrying HLA-A2+DQβ0603:DQα0103 but negative with the targets carrying HLA-A1/A3+DQβ0603:DQα0103. In our experience, DQβ0603:DQα0103 antibodies do not bind to cells despite MFI. If they do not bind to cells, the downstream function and clinical utility of DQβ0603:DQα0103 are questionable. Therefore, please provide cell-binding data and possibly transplants showing relevance in ABMR.

Yes, we have conducted Flow XM on all six sera using B cells from DQβ0603:DQα0103 positive donors. They were all considered negative in B cell XM. We have rationalized that the bead positive, XM negative was most likely due to the lower sensitivity of the cell-base assay. The expression of DQ molecules on the cell surface is significantly lower than that of DR molecules¹⁴ and if only a fraction of those DQ molecules would have the correct peptide recognizable by the antibodies, it would further reduce the number of DQ molecules that could be recognized by the peptide-specific DQβ0603:DQα0103 antibodies. To further demonstrate the low abundance of DQ molecules in the total

Class II molecules (DR, DQ and DP) on the B cells, we purified the HLA Class II proteins from the five B cell lines used in this study with either Class II specific or DQ specific affinity columns. All six sera with DQ β 0603:DQ α 0103 antibody reacted very weakly or negatively with beads presenting total Class II proteins (blue bar), but strongly positive with beads presenting the DQ fraction (orange bar) from the same B cell lines (Figure a below). The reactivity of the DQ-specific mAb, FM5148, was weaker in total Class II antigens beads compared to DQ only beads in contrast to the anti-Class II mAb, which showed similar reactivity between both types of beads (Figure b below). These data support the proposition that cell-based assay has lower sensitivity than the bead-based assay.

- Fig.2: Please provide the control reactivities (i.e., DQ6-PC and DQ α /DQ β mAbs) of recombinant DQ β 0603:DQ α 0103 proteins extracted from six stable transfectants of CD74 and DM positive host cell lines. What is the impact of HLA class I genotypes on these reactivities? Why is S6 completely negative with all 6 beads?

Control sera and mAb reactivities were added to Figure 2b. The possible causes of negative reactivity of S6 to all six beads were added to the end of the section, see Line 168-171. Positive reactivity of S6 was restored by the expression of HLA C*16:01 in LCLKO host cell line.

Minor Comments:

- What was the purity and yield of peptides? The HPLC report is missing in the manuscript. Electrospray Ionization (ESI) histograms that generally show the presence of peptides detected by mass spectrometry are also missing.

The peptides used in this study were all >95% purity. The HPLC report and mass spectrometry histogram for all peptides used are provided in supplemental files.

2. It is confusing to see the amino acid positions noted using “codon,” which is meant for the nucleotide sequence.

The incorrectly used terminology of codon has been replaced with amino acid position in the text.

3. Page 15, line no. 264: replace “demostrated” with “demonstrated.”

Corrected, thank you.

4. The references are not in one format. For example, some references have a full stop at the end but others do not.

Corrected, thank you.

REVIEWER COMMENTS

Reviewer #2 (expert in HLA immunogenetics and structure):

The authors have considered reviewers' recommendations and have performed additional work and revisions of the manuscript. In my opinion, they have convincingly showed that Ab reactivity is influenced by the differential presentation of peptides as presented by HLA-DQ molecule DQ β *0603:DQ α *0103. Understandably, they cannot address all aspects surrounding the complex interactions taking place among HLA molecule-bound peptides and Abs but they have made a sincere effort to address most of our comments. Overall, the manuscript has improved significantly.

Reviewer #3 (expert in immunogenetics and transplantation):

Thank you for providing me with the opportunity to review the revised manuscript. The authors have not addressed three of my significant comments, which are listed below. These concerns can be resolved through additional experiments and discussed within the limitations section.

My original comment#3: The data presented here indicate that DQ β 0603:DQ α 0103 molecules present peptides of the cognate HLA-A2 heavy-chain, expressed on the same EBV-transformed B cell lines. Presenting self-HLA class I heavy-chain-derived peptides by DQ molecules challenges the current dogma.

Does this mean that healthy individuals having DQ β 0603:DQ α 0103 and HLA-A2 should be generating antibodies against DQ β 0603:DQ α 0103? Do you have data supporting this?

The authors' response was, "We don't know whether healthy individuals will produce this type of antibody or not since we have not investigated this", which is inadequate response. The authors supervise HLA laboratories and likely have tested hundreds of patients with DQ β 0603:DQ α 0103 and HLA-A2 genotypes for HLA antibodies. Therefore, they should provide this data. I believe that none of these patients will produce antibodies against DQ β 0603:DQ α 0103, even if they possess the DQ β 0603:DQ α 0103 and HLA-A2 genotypes. Consequently, this data could potentially contradict the current findings.

My original comment#4: Table 2 lists HLA class I peptides identified from the DQ β 0603:DQ α 0103 protein peptidome analysis. The HLA-A*02:01 heavy-chain peptide carrying the epitope (aa 142TTKH145) is identified as the determinant of the DQ β 0603:DQ α 0103 antibody specificity in Serum S1. This epitope aa-142TTKH145 is also shared by A68 and A69. Do you have data from cell lines expressing DQ β 0603:DQ α 0103 and HLA-A68/A69?

The authors' response was, "Unfortunately, we don't have DQ β 0603:DQ α 0103 and A*68/*69 co-expressing cell line in our collection". The 10th International Histocompatibility Workshop cell line panel should include this B cell line, and the authors should conduct this control experiment to determine whether the epitope aa-142TTKH145 is indeed influencing binding specificity.

My original comment#5: In the introduction, the authors clearly outlined the gap in the field - the inconsistency between solid-phase assay antibody results and cell-based cross-matches, causing inefficiency in organ allocation and ABMR diagnosis/treatment. However, the present study completely ignored such fundamental experiments of cell-binding assays. Have you performed a cell-based crossmatch of all 6 sera with DQ β 0603:DQ α 0103 antibody? In particular, to confirm serum#1 reactivity in Fig.1a, it will be interesting to see a positive crossmatch with a target carrying HLA-A2+

DQ β 0603:DQ α 0103 but negative with the targets carrying HLA-A1/A3+DQ β 0603:DQ α 0103. In our experience, DQ β 0603:DQ α 0103 antibodies do not bind to cells despite MFI. If they do not bind to cells, the downstream function and clinical utility of DQ β 0603:DQ α 0103 are questionable. Therefore, please provide cell-binding data and possibly transplants showing relevance in ABMR.

The authors' response was, "Yes, we have conducted Flow XM on all six sera using B cells from DQ β 0603:DQ α 0103 positive donors. They were all considered negative in B cell XM. We have rationalized that the bead positive, XM negative was most likely due to the lower sensitivity of the

cell-base assay". I find it challenging to comprehend the clinical significance of antibodies that fail to bind to the cells. The authors should elucidate, with clinical evidence, how a patient can possess donor-specific DQ β 0603:DQ α 0103 antibodies, test negative in the B cell crossmatch, and yet experience allograft rejection. Without such clinical evidence, the conclusions of this paper hold no value and may create confusion within the transplant immunology community.

RESPONSE TO REVIEWERS' COMMENTS

Authors' responses in green

Reviewer #2 (expert in HLA immunogenetics and structure):

The authors have considered reviewers' recommendations and have performed additional work and revisions of the manuscript. In my opinion, they have convincingly showed that Ab reactivity is influenced by the differential presentation of peptides as presented by HLA-DQ molecule DQ β *0603:DQ α *0103. Understandably, they cannot address all aspects surrounding the complex interactions taking place among HLA molecule-bound peptides and Abs but they have made a sincere effort to address most of our comments. Overall, the manuscript has improved significantly.

We thank Reviewer #2 for the support of publication for our manuscript. We agree that the additional experiments and discussion based on the constructive comments have improved the manuscript significantly.

Reviewer #3 (expert in immunogenetics and transplantation):

Thank you for providing me with the opportunity to review the revised manuscript. The authors have not addressed three of my significant comments, which are listed below. These concerns can be resolved through additional experiments and discussed within the limitations section.

My original comment#3: The data presented here indicate that DQ β 0603:DQ α 0103 molecules present peptides of the cognate HLA-A2 heavy-chain, expressed on the same EBV-transformed B cell lines. Presenting self-HLA class I heavy-chain-derived peptides by DQ molecules

challenges the current dogma.

Does this mean that healthy individuals having DQ β 0603:DQ α 0103 and HLA-A2 should be generating antibodies against DQ β 0603:DQ α 0103? Do you have data supporting this?

The authors' response was, "We don't know whether healthy individuals will produce this type of antibody or not since we have not investigated this", which is inadequate response. The authors supervise HLA laboratories and likely have tested hundreds of patients with DQ β 0603:DQ α 0103 and HLA-A2 genotypes for HLA antibodies. Therefore, they should provide this data. I believe that none of these patients will produce antibodies against DQ β 0603:DQ α 0103, even if they possess the DQ β 0603:DQ α 0103 and HLA-A2 genotypes. Consequently, this data could potentially contradict the current findings.

We thank Reviewer #3 for the continued detailed review of our manuscript. We believe that there might be a slight misunderstanding with respect to this specific question. We have presented data, which demonstrates that anti-HLA antibodies detected in some patient in the setting of solid organ transplantation can recognize an epitope on a DQ β 0603:DQ α 0103 protein in association with HLA Class I peptides derived from the same EBV-transformed B cells or transfectants. We have not suggested or discussed that these DQ β 0603:DQ α 0103 antibodies were generated in patients who also possess the DQB1*06:03/DQA1*01:03 and HLA-A*02 genotype. We have searched the database of one of the authors' transplant centers that has over 20,000 patients on record as Reviewer #3 has suggested. There are less than a hundred patients in the database possessing the specified HLA genotype. Around 10% of the patients that have this genotype have detectable anti-HLA antibodies and none of them has the DQ β 0603:DQ α 0103 antibody. Furthermore, none of the 22 cases with antibody reactivity towards DQ β 0603:DQ α 0103

presented in our study were from patients carrying the DQB1*06:03/DQA1*01:03 genotype. We have added this information in the discussion of the adapted manuscript to further clarify this point (line 303).

While this is an interesting question asked by Reviewer #3 and one that could be potentially investigated in subsequent studies, we feel the presentation of self-peptides on DQ molecules derived from class I does not have to imply that antibodies have to be made against this constellation in healthy individuals. Whether such antibodies can or cannot be detected in healthy carriers of the designated HLA types does not in our opinion present a contradiction to, nor detract from our findings. Our search in a large dataset of patients in the setting of organ transplantation (not healthy individuals) did not generate any such cases, which suggests that this would perhaps be a rare event.

My original comment#4: Table 2 lists HLA class I peptides identified from the DQβ0603:DQα0103 protein peptidome analysis. The HLA-A*02:01 heavy-chain peptide carrying the epitope (aa 142TTKH145) is identified as the determinant of the DQβ0603:DQα0103 antibody specificity in Serum S1. This epitope aa-142TTKH145 is also shared by A68 and A69. Do you have data from cell lines expressing DQβ0603:DQα0103 and HLA-A68/A69?

The authors' response was, "Unfortunately, we don't have DQβ0603:DQα0103 and A*68/*69 co-expressing cell line in our collection". The 10th International Histocompatibility Workshop cell line panel should include this B cell line, and the authors should conduct this control experiment to determine whether the epitope aa-142TTKH145 is indeed influencing binding specificity.

We have searched the 10th International Histocompatibility Workshop cell line panel before our initial response and did not find any homozygous DQB1*06:03/DQA1*01:03 cell line with A*68 or A*69 genotype and we have reconfirmed this with a second search.

DQB1*06:03/DQA1*01:03 associated with either A*68 or A*69 is not a common haplotype^{1,2}, so it's very difficult to find cells with such genotype. Using a homozygous

DQB1*06:03/DQA1*01:03 cell line with A*68 or A*69 is essential to perform the experiment Reviewer #3 requested because if a heterozygous DQ cell line is used, the DQ proteins from the non-DQB1*06:03/DQA1*01:03 allele and the trans-heterodimers formed between the two different DQB1 and DQA1 alleles (this was also mentioned by Reviewer #1 in his comments) will interfere with the interpretation of the data.

Nevertheless, we believe our data showing that the transfection of A*02:01, but not C*16:01, reestablished the reactivity of serum S1 in Figure 3 with the supporting data from peptide loading experiment in Figure 4 are sufficient to demonstrate the peptide sequence specificity of the S1 epitope. Additionally, the completely opposite reactivity profile of serum S6 from Figure 3 and 4 where C*16:01, but not A*02:01, reestablished the serum S6 reactivity supports our finding that amino acid substitutions in the peptides can influence the antibody binding specificity.

My original comment#5: In the introduction, the authors clearly outlined the gap in the field - the inconsistency between solid-phase assay antibody results and cell-based cross-matches, causing inefficiency in organ allocation and ABMR diagnosis/treatment. However, the present study completely ignored such fundamental experiments of cell-binding assays. Have you performed a cell-based crossmatch of all 6 sera with DQβ0603:DQα0103 antibody? In particular, to confirm

serum#1 reactivity in Fig.1a, it will be interesting to see a positive crossmatch with a target carrying HLAA2+

DQβ0603:DQα0103 but negative with the targets carrying HLA-A1/A3+DQβ0603:DQα0103.

In our experience, DQβ0603:DQα0103 antibodies do not bind to cells despite MFI. If they do not bind to cells, the downstream function and clinical utility of DQβ0603:DQα0103 are questionable. Therefore, please provide cell-binding data and possibly transplants showing relevance in ABMR.

The authors' response was, "Yes, we have conducted Flow XM on all six sera using B cells from DQβ0603:DQα0103 positive donors. They were all considered negative in B cell XM. We have rationalized that the bead positive, XM negative was most likely due to the lower sensitivity of the cell-based assay". I find it challenging to comprehend the clinical significance of antibodies that fail to bind to the cells. The authors should elucidate, with clinical evidence, how a patient can possess donor-specific DQβ0603:DQα0103 antibodies, test negative in the B cell crossmatch, and yet experience allograft rejection. Without such clinical evidence, the conclusions of this paper hold no value and may create confusion within the transplant immunology community.

We agree with Reviewer #3 that the clinical implications of our data are important however we do not think that our negative B cell crossmatch results should be interpreted in a way that our detected DQβ0603:DQα0103 reactive antibodies are not binding to their target in these assays. Instead as mentioned in our previous response this is most likely due to the low expression of DQ molecules on the B cells. Even though we were unable to show positive flow cross-match results from S1-S6 using B cells with DQB1*06:03/DQA1*01:03 genotype isolated from blood,

we provided additional data on the cell-based flow cytometric assay (Figure 2c and 2d). These data demonstrate that S1 could react to transfectants expressing DQ β 0603:DQ α 0103 and A0201 proteins on the cell surface supporting the suggestion that our results are related to expression of the DQ alleles on the cell surface and not antibody binding. It is well known clinically that anti-DQ antibodies are difficult to detect with the Flow cross-match (based on low DQ expression on B-cells) in general. There is also literature that show the detrimental effect of pre-transplant DSA detected with the SAB where sensitive cell based cross-match techniques such as the Flow based cross-match is negative and that this is especially pronounced for Class II directed DSA^{3,4,5}. The detection of pre-transplant or de novo DSA directed against HLA-DQ has also previously been associated with particularly poor outcomes in kidney transplantation, which further suggests that our findings may be clinically important^{6,7,8}. We have also in the revised manuscript presented data from an additional case, S22, that showed the appearance of a *de novo* DQ β 0603:DQ α 0103 DSA post-transplantation, with a similar peptide dependent reactivity pattern, coinciding with the diagnosis of AMR, which further support the potential clinical relevance of such an antibody. In order to clarify this important point, we have adapted the text and references in the discussion section of the revised manuscript (line 353).

We appreciate the detailed review by Reviewer #3 on the remaining three concerns and the possibility it presents to us to better address these important concerns. We hope that our responses have addressed the remaining concerns from Reviewer #3 and that our adaptations have improved the clarity of our manuscript.

References:

1. Maiers, M., Gragert, L. & Klitz, W. High-resolution HLA alleles and haplotypes in the United States population. *Hum Immunol.* **68**, 779-788 (2007).
2. Gragert, L., Madbouly, A., Freeman, J. & Maiers, M. Six-locus high resolution HLA haplotype frequencies derived from mixed-resolution DNA typing for the entire US donor registry. *Hum Immunol.* **74**: 1313-1320 (2013),
3. Kwon, H. et al. Impact of pretransplant donor-specific antibodies on kidney allograft recipients with negative flow cytometry cross-matches. *Clin Transplant.* **32**:e13266. doi:10.1111/ctr.13266 (2018).
4. Adebiyi, O. O. et al. Clinical Significance of Pretransplant Donor-Specific Antibodies in the Setting of Negative Cell-Based Flow Cytometry Crossmatching in Kidney Transplant Recipients. *Am J Transplant.* **16**, 3458-3467 (2016).
5. Patel, A. M., Pancoska, C., Mulgaonkar, S. & Weng, F. L. Renal transplantation in patients with pre-transplant donor-specific antibodies and negative flow cytometry crossmatches. *Am J Transplant.* **7**, 2371-2377 (2007).
6. Frischknecht, L. et al. The impact of pre-transplant donor specific antibodies on the outcome of kidney transplantation - Data from the Swiss transplant cohort study. *Front. Immunol.* **13**:1005790. doi: 10.3389/fimmu.2022.1005790. eCollection (2022).
7. DeVos, J. M. et al. Donor-specific HLA-DQ antibodies may contribute to poor graft outcome after renal transplantation. *Kidney Int.* **82**, 598-604 (2012).
8. Senev, A. et al. Specificity, strength, and evolution of pretransplant donor-specific HLA antibodies determine outcome after kidney transplantation. *Am J Transplant.* **19**, 3100-3113 (2019).

REVIEWERS' COMMENTS

Reviewer #3 (expert in immunogenetics and transplantation):

Thanks for addressing my comments. Greatly appreciated.

RESPONSE TO REVIEWERS' COMMENTS

Authors' responses in green

Reviewer #3 (expert in immunogenetics and transplantation):

Thanks for addressing my comments. Greatly appreciated.

We appreciate the constructive comments from Reviewer #3. We are glad that we were able to address all the concerns raised by Reviewer #3.